# Improving the Spatial Characteristics of Three-Level LUT-Based Mealy FSM Circuits

Alexander Barkalov [1,2,*,†], Larysa Titarenko [1,3,†], Małgorzata Mazurkiewicz [4,*,†] and Kazimierz Krzywicki [5,†]

1 Institute of Metrology, Electronics and Computer Science, University of Zielona Góra, ul. Licealna 9, 65-417 Zielona Góra, Poland

2 Department of Computer Science and Information Technology, Vasyl Stus' Donetsk National University, 600-richya Str. 21, 21021 Vinnytsia, Ukraine

3 Department of Infocommunication Engineering, Faculty of Infocommunications, Kharkiv National University of Radio Electronics, Nauky Avenue 14, 61166 Kharkiv, Ukraine

4 Institute of Control & Computation Engineering, University of Zielona Góra, ul. Licealna 9, 65-417 Zielona Góra, Poland

5 Department of Technology, The Jacob of Paradies University, ul. Teatralna 25, 66-400 Gorzów Wielkopolski, Poland

* Correspondence: a.barkalov@imei.uz.zgora.pl (A.B.); m.mazurkiewicz@issi.uz.zgora.pl (M.M.)

† These authors contributed equally to this work.

**Abstract:** The main purpose of the method proposed in this article is to reduce the number of look-up-table (LUT) elements in logic circuits of sequential devices. The devices are represented by models of Mealy finite state machines (FSMs). Thesee are so-called MPY FSMs based on two methods of structural decomposition (the replacement of inputs and encoding of output collections). The main idea is to use two types of state codes for implementing systems of partial Boolean functions. Some functions are based on maximum binary codes; other functions depend on extended state codes. The reduction in LUT counts is based on using the method of twofold state assignment. The proposed method makes it possible to obtain FPGA-based FSM circuits with four logic levels. Only one LUT is required to implement the circuit corresponding to any partial function. An example of FSM synthesis using the proposed method is shown. The results of the conducted experiments show that the proposed approach produces LUT-based FSM circuits with better area-temporal characteristics than for circuits produced using such methods as Auto and One-hot of Vivado, JEDI, and MPY FSMs. Compared to MPY FSMs, the values of LUT counts are improved. On average, this improvement is 8.98%, but the gain reaches 13.65% for fairly complex FSMs. The maximum operating frequency is slightly improved as compared with the circuits of MPY FSMs (up to 0.64%). For both LUT counts and frequency, the gain increases together with the growth for the numbers of FSM inputs, outputs and states.

**Keywords:** Mealy FSM; FPGA; LUT; synthesis; replacement of inputs; collections of outputs; twofold state assignment

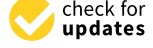



## 1. Introduction

To represent various sequential blocks, a model of a Mealy finite state machine (FSM) [1] can be applied. There are many examples of using this model in the implementation of various digital systems [2]. In this paper, we consider FSM circuits implemented using field-programmable gate arrays (FPGAs) [3,4]. This choice is due to the wide use of FPGAs in the implementation of a wide variety of projects [4,5]. Leading experts are confident that FPGAs will continue to dominate logic design for at least the next twenty years [6].

When using any logic basis for the implementation of FSM circuits, a number of optimization problems always arise [7,8]. One of the most important tasks is to obtain a circuit

that is optimal in terms of hardware costs. By optimal, we mean a circuit that consumes the minimum possible amount of chip resources while simultaneously providing the required level of performance and power consumption. In the case of FPGA-based circuits [9], the optimization strategy significantly depends on the types of configurable logic blocks (CLBs) used [10]. In this paper, we discuss the most common CLBs which include look-up table (LUT) elements, programmable flip-flops, and dedicated multiplexers [10,11]. To combine these CLBs into an FSM circuit, the following chip resources are used: the synchronization tree, programmable interconnections, and programmable input-outputs [12,13]. The method proposed in this paper is aimed at reducing the number of LUTs (LUT count) in a resulting FSM circuit.

It is generally accepted that reducing LUT count leads to improving the spatial characteristics of FSM circuits (reducing the occupied chip areas) [14,15]. Area reduction can be achieved by applying structural decomposition (SD) methods [9] leading to multilevel FSM circuits. However, such a reduction may have an overhead [9]. This overhead consists of a significant performance degradation compared to equivalent single-level FSM circuits [14,16]. However, performance has to be sacrificed if the criterion of design optimality is the minimum occupied chip area.

The best LUT counts can be obtained for three-level FSM circuits when the methods of replacing FSM inputs and encoding collections of FSM outputs [17] are used together. However, for sufficiently complex FSMs, some of the logic blocks (or even all three blocks) may have a multilevel structure. This leads to an increase in the number of logical levels and interconnections. In turn, this leads to an increase in the occupied area, power consumption and delay time of the FSM circuit. In this paper, we propose a method to reduce the LUT counts of three-level FSM circuits. The proposed method is based on using twofold state assignment [18]. This approach leads to a decrease in the number of LUTs and their levels in the resulting LUT-based FSM circuits.

There are some leading companies producing FPGA chips. The largest producer is AMD Xilinx [19]. As follows from [4], FPGAs from AMD Xilinx are widely used in various projects. Due to this, we structured our approach according to the FPGA families [19] by AMD Xilinx. In our research, we use FPGAs from the VIrtex-7 family [10].

The article contains several new scientific results. Firstly, a new architecture of an LUT-based Mealy FSM circuit is proposed. Secondly, methods for the uniform distribution of inputs and state encoding are proposed, which make it possible to reduce the number of LUTs in the circuit of the input replacement block in comparison with the known methods for implementing this block. Thirdly, a new method for stabilizing FSM outputs is proposed, in which the input register is replaced by a register of output collection codes. The noted new approaches led to the main contribution of the article, which is a novel design method aimed at hardware reduction in the multilevel circuits of LUT-based Mealy FSMs. The hardware reduction is achieved due to the use of two types of state codes. The maximum binary state codes are used to replace the FSM inputs. Other partial Boolean functions depend on extended state codes. The proposed approach leads to four-level FSM circuits where any partial function is represented by a single LUT. The conducted experiments show that the resulting FSM circuits include fewer LUTs compared to equivalent three-level circuits [17]. It is very important that the hardware reduction does not lead to the significant deterioration of temporal characteristics.

The rest of the paper is organized as follows. Section 2 shows the peculiarities of the LUT-based Mealy FSM design. The analysis of related works is discussed in Section 3. Section 4 presents the main idea of our method. In Section 5, we include a step-by-step example showing how to apply the proposed method. Section 6 includes the experimental results. The last part of the article is a short conclusion.

## 2. Peculiarities of LUT-Based Mealy FSM Design

The law of the behaviour of a Mealy FSM can be represented using three sets and two functions [20]. These sets are the following: a set of internal states $S = \{s_1, \ldots, s_M\}$, a set of

inputs $X = \{x_1, \ldots, x_L\}$, and a set of outputs $Y = \{y_1, \ldots, y_N\}$. The interstate transitions are represented by a function of transitions. An output function shows the FSM outputs generated during these transitions. In this article, we use a state transition graph (STG) [1] as an initial tool for FSM design. An STG consists of vertices representing FSM states. The vertices are connected by arcs corresponding to interstate transitions. Each arc is marked by an input signal (the conjunction of inputs leading to a particular transition) and a collection of outputs associated with this transition [1]. To synthesize the FSM circuit, we transformed this STG into the equivalent state transition table (STT) [1].

To design an FSM circuit, it is necessary to replace abstract states $s_m \in S$ with binary codes $K(s_m)$. This is the state-assignment step [1]. To minimize the number of state variables and input memory functions (IMFs), it is necessary to minimize the bitness of state codes. The minimum possible number $R_{MB}$ of state-code bits corresponds to a maximum state assignment [20]. This number is determined as

$$R_{MB} = \lceil \log_2 M \rceil. \tag{1}$$

To encode states, state variables creating a set $T = \{T_1, \ldots, T_{RMB}\}$ are used. To keep the state codes, a special register, RG, consisting of $R_{MB}$ flip-flops is used as a part of FSM circuit.

In most practical cases [9], as elements of the state register are used the synchronous D flip-flops. Each state variable is represented by a unique flip-flop. The input of the $r$-th flip-flop is connected with an IMF $D_r \in D$ where $D = \{D_1, \ldots, D_{RMB}\}$ is a set of IMFs. The initial state code is forcibly loaded into RG. To do this, a special pulse of initialization *Start* is used. Set D determines a state code loaded into RG. To load a code $K(s_m)$, the pulse of synchronization *Clock* is used.

Using either STG or STT, a direct structure table (DST) [20] can be constructed. There are six columns in the DST [20]: $s_C$, $s_T$, $X_h$, $Y_h$, $D_h$, $h$. The data from these columns have the following meaning: $s_C$ is an initial state for a given transition; $s_T$ is a final state for this transition; $X_h$ is a conjunction of FSM inputs determining the transition $\langle s_C, s_T \rangle$; $Y_h$ is a collection of outputs (CO) produced during the transition $\langle s_C, s_T \rangle$; $D_h$ is a set of IMFs equal to 1 to execute the $h$-th transition (to load the code $K(s_T)$ into RG); and $h$ is the transition number ($h \in \{1, \ldots, H\}$). The DST is a base for constructing the following systems of Boolean functions (SBFs) [21]:

$$D = D(T, X); \tag{2}$$

$$Y = Y(T, X). \tag{3}$$

The SBFs (2) and (3) are a base for implementing the so-called P Mealy FSM [9]. In FPGA-based FSMs, the flip-flops of RG are distributed among the CLBs, including LUTs, generating the functions (2). Thus, the distributed state-code register is hidden. As a result, there are only two blocks in the structural diagram of LUT-based P Mealy FSM (Figure 1).

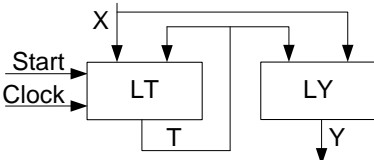

**Figure 1.** Structural diagram of LUT-based P Mealy FSM.

The LUTs of a block LT implement IMFs (2). The memory elements of LT create the RG. This explains why the pulses *Start* and *Clock* enter LT. Obviously, the state variables $T_r \in T$ come out of the block LT. The block LY generates functions (3) representing the outputs $y_n \in Y$. Each LUT has $S_L$ inputs.

The functions (2) and (3) are represented by their sum of products (SOPs) [1]. An SOP of a Boolean function $f_i \in D \cup Y$ has $NI(f_i)$ literals. For rather complex FSMs, the following condition may hold:

$$NI(f_i) > S_L. \tag{4}$$

If (4) takes place, then the circuit of P Mealy FSM is multi-level. It is known [9] that multi-level circuits are less efficient than the equivalent single-level circuits (the former are much slower and require more power than the latter). The same is true for the numbers of interconnections in the equivalent single-level and multi-level circuits. The growth in interconnections leads to the further growth in the values of both time of cycle and power consumption. The use of SD-based methods can lead to a significant improvement in the overall circuit quality [9,17].

There are two types of literals in SOPs of functions (2) and (3): external inputs $x_l \in X$ and elements of the set $T$ (the variables $T_r \in T$). Each function $f_i \in D \cup Y$ depends on $R_i \leq R_{MB}$ state variables and $L_i \leq L$ inputs. There is only one LUT in the circuit corresponding to the function $f_i \in D \cup Y$, if the following condition is true:

$$R_i + L_i \leq S_L. \tag{5}$$

If condition (5) holds, then the values of function $f_i \in D \cup Y$ are generated by a single-LUT circuit. If condition (5) takes place for all $R + N$ functions, then the circuit of P Mealy FSM is single-level. A single-level circuit has the best possible values of the required chip area, power consumption and maximum operating frequency.

However, there are FSMs with around 500 states and 30 inputs [2]. In this case, each function $f_i \in D \cup Y$ may depend on up to 39 arguments. Thus, their SOPs can include up to 39 literals. Of course, these SOPs cannot be implemented using only a single LUT with $S_L = 6$ inputs. Thus, the corresponding circuits will be multi-level with spaghetti-type interconnecting systems. To improve the characteristics of multi-level circuits, various optimization methods should be applied. In this paper, we propose an approach which allows reducing the chip area occupied by the LUT-based FSM circuit when the condition (5) is violated.

## 3. Brief Analysis of Related Works

The problem of area reduction is discussed in thousands of monographs and articles. For example, various methods for solving this problem are proposed in the following works (to name but a few): [14,22–28]. As follows from [23], reducing the required chip area is connected with reducing the LUT count for a corresponding circuit. To achieve this goal, three groups of methods can be used: a proper state assignment, a functional decomposition (FD) of Boolean functions, and SD-based approaches [9].

The proper state assignment leads to the elimination of some literals from SOPs (2) and (3) [20]. If the elimination of literals results in the fulfilment of condition (5) for SOPs of all functions (2) and (3), then the resulting FSM circuit is single-level. This can be achieved using, for example, the state assignment method JEDI distributed with the CAD system SIS [29]. JEDI-based optimization is achieved by creating adjacent codes for states whose transitions depend on the same FSM inputs $x_l \in X$. As shown in [30], this allows elimination of up to 3 literals from SOPs representing benchmark FSMs from the library LGSynth93 [31]. Thus, JEDI can solve the optimization problem if the relation $NI(f_i) - S_L \leq 3$ holds. However, this relation only takes place for rather simple FSMs [9].

As follows from various research [32–35], there is no best universal state-assignment approach. For example, optimization success depends on how many variables $x_l \in X$ the transitions from each state depend on. For different FSMs, the same state-assignment method may either improve or deteriorate the quality of resulting circuits. In addition, the optimization strategy depends strongly on the peculiarities of the logic elements used [33]. If LUTs are used, the spatial improvement can be achieved due to an increase in the state-code length [36]. In the extreme case, the number of bits is equal to $M$. This is a one-hot

state assignment [1], when the RG includes $M$ flip-flops. The results of research reported in [32] show that the one-hot state assignment can improve the FSM characteristics, if there is $M > 16$. However, it is necessary to take into account the number of FSM inputs [34]. As shown in [32], using MBC improves FSM quality if there is $L > 10$ (compared to FSMs with one-hot codes). This situation stimulates the development of new types of state codes and encoding strategies.

If no state-assignment method allows the implementation of a single-level circuit for a given FSM, then decomposition methods should be applied. In this case, the initial functions (2) and (3) are represented as a composition of partial Boolean functions (PBFs). The decomposition is executed till the condition (4) is satisfied for each partial function. Any kind of decomposition leads to a multi-level FSM circuit.

In the case of FD-based FSM circuits, CLBs are connected by complicated systems of "spaghetti-type" interconnections [11]. Such circuits have much lower clock rates compared to equivalent single-level solutions. This is connected with the fact that, now, "...wires delay has come to dominate logic delay" [37]. In addition, compared to single-level circuits, FD-based circuits are more power-consuming. This phenomenon is due to the fact that the interconnections absorb up to 70% of the total power consumed by an FPGA-based FSM circuit [37]. However, the advantage of FD is that it is applicable to the implementation of Boolean functions of any practical complexity. Therefore, FD-based algorithms are used in all industrial CAD systems aimed at the implementation of FPGA-based digital systems [38–41].

In many cases, the methods of structural decomposition [9] allow the production of FSM circuits with better space-time-energy characteristics than their FD-based counterparts. The SD-based FSM circuits can be viewed as a composition of large logic blocks with unique input-output systems. Such an approach leads to the regularization of interconnections compared to FD-based FSM circuits [16]. Different methods of SD can be used together. Due to this, the number of blocks can vary from 2 to 4, depending on how many methods are used. The methods of SD and FD can be used together [9].

Two methods of SD are most commonly used. One of them is the replacement of inputs (RI) with some additional variables [9]. The second method is the encoding of COs [9]. Below is a brief description of these methods.

The process of RI comes down to replacing inputs $x_l \in X$ with the additional variables from a set $B = \{b_1, \ldots, b_G\}$. The replacement makes sense if $L \gg G$ [9]. As a result, the SBFs (2) and (3) are replaced by the systems

$$B = B(T, X); \tag{6}$$

$$D = D(T, B); \tag{7}$$

$$Y = Y(T, B). \tag{8}$$

The system (6) is represented by a block with inputs $x_l \in X$ and $T_r \in T$. In the following text, we denote this block with the symbol LB. Obviously, the circuit of LB consumes some chip resources. The systems (7) and (8) are implemented by block LTY. This approach makes sense if the SOPs (7) and (8) include much fewer literals than the SOPs (2) and (3) [9]. In this case, the LUT counts in the circuit of P FSM significantly exceed the total number of LUTs necessary to implement SBFs (6)–(8).

During the interstate transitions, Q different COs $Y_q \subseteq Y$ are generated. Each CO can be represented by a code $K(Y_q)$. This code includes $R_{CO}$ bits [9]:

$$R_{CO} = \lceil \log_2 Q \rceil. \tag{9}$$

The COs are encoded using some additional variables creating a set $Z = \{z_1, \ldots, z_{RCO}\}$. If this approach is applied together with the RI, then the SBF (3) is replaced with the following SBFs:

$$Z = Z(T, B); \tag{10}$$

$$Y = Y(Z). \tag{11}$$

The system (10) depends on the same variables as the system (7). Thus, these two SBFs are implemented using the same block, LTZ. To implement SBF (11), block LY is used. Sharing these methods turns the original P FSM (Figure 1) into MPY FSM (Figure 2).

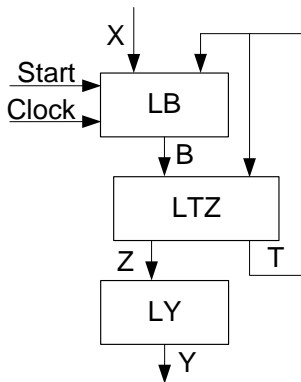

**Figure 2.** Structural diagram of MPY Mealy FSM.

In MPY FSM, the block LB generates the additional variables (6). The block LTZ generates IMFs represented by (7) and additional variables (10). The block LY generates the FSM outputs (11). As shown in [17], the transition from P FSM to MPY FSM allows the reduction in LUT counts in equivalent FSM circuits. Of course, this area reduction leads to a decrease in the value of maximum operating frequency. This decrease can be viewed as the area-reducing overhead.

To obtain SBF (6), a table of RI should be constructed [20]. Its columns are marked by states $s_m \in S$, whereas additional variables $b_g \in B$ mark its rows. There is a symbol $x_l$ written at the intersection of a row $b_g \in B$ and column $s_m \in S$, if the variable $b_g \in B$ replaces the input $x_l \in X$ for the state $s_m \in S$. In fact, the block LB is a multiplexer, the information inputs of which are connected to inputs $x_l \in X$ and the control inputs are connected to state variables $T_r \in T$.

To obtain SBFs (7) and (10), it is necessary to create a transformed DST. In the transformed DST, the column $X_h$ is replaced by a column $B_h$, whereas the column $Y_h$ is replaced by a column $Z_h$. These new columns are filled in as follows. For example, the first row of DST includes a CO $Y_2$ generated during a transition $\langle s_1, s_2 \rangle$ caused by the input signal $X_1 = x_1 x_2$. Let the following relations take places for the state $s_1 \in S : x_1 = b_1$ and $x_2 = b_2$. In this case, the input signal $X_1 = x_1 x_2$ is replaced by the conjunction $B_1 = b_1 b_2$ written in the column $B_h$. If $K(Y_2) = 101$, then the additional variables $z_1, z_3 \in Z$ are written in the column $Z_h$. All other rows of the transformed DST are filled in the same manner.

To obtain SBF (11), it is necessary to create the Karnaugh map whose cells are marked by the variables $z_r \in Z$. The symbols $Y_q$ are written inside the cells. Using this map, the minimized SOPs (11) are constructed. The minimization makes sense if some literals are eliminated from all product terms of a SOP representing a function $y_n \in Y$ [9].

The application of this approach is most efficient if condition (4) is satisfied for all functions $f_i \in B \cup D \cup Z \cup Y$ [9]. Otherwise, there will be more than a single LUT in the circuits for functions that do not satisfy condition (4). Moreover, this leads to the multi-levelness of the corresponding blocks, which further reduces the MPY FSM performance. To implement these multi-level circuits, the methods of FD should be applied.

To overcome this shortcoming of MPY FSM, we propose to transform its structural diagram using the method of two-fold state assignment (TSA) [18]. This idea is discussed in the next section.

## 4. Main Idea of Proposed Method

To execute the TSA, it is necessary to create a partition $\pi_S = \{S^1, \ldots, S^K\}$ of the set of states. As a result, each state $s_m \in S$ has two codes. The maximum binary code $K(s_m)$ has $R_{MB}$ bits. This code represents a state as some element of the set $S$. The partial code $C(s_m)$ represents a state as some element of a class $S^k \in \pi_S$. This class includes $M_k$ elements. To encode them, $R_k$ bits are sufficient:

$$R_k = \lceil \log_2(M_k + 1) \rceil. \tag{12}$$

In (12), the value of $M_k$ is incremented to encode the relation $s_m \notin S^k$. We use the code with all zeroes to encode this relation. This code represents the state $s_m \in S^k$ for all classes other than $S^k$.

The codes $C(s_m)$ for all classes $S^k \in \pi_S$ form an extended state (ESC) code of the state $s_m \in S^k$. Each ESC includes $R_S$ bits, where

$$R_S = R_1 + \cdots + R_K. \tag{13}$$

To create ESCs, the additional variables are used. These variables are elements of a set $V = V^1 \cup V^2 \cup \ldots \cup V^K$. The variables $v_r \in V^k$ create the codes $C(s_m)$ for the states $s_m \in S^k$. To generate ESCs, it is necessary to transform state codes $K(s_m)$ into codes $C(s_m)$ for all states $s_m \in S$. To transform the codes, it is necessary to create the following SBF:

$$V = V(T). \tag{14}$$

We discuss a case wherein both the replacement of inputs and encoding of COs are executed. In this case, each class $S^k \in \pi_S$ determines three sets. A set $B^k \subseteq B$ includes variables $b_g \in B$ determining transitions from the states $s_m \in S^k$. A set of additional variables $Z^k \subseteq Z$ includes elements determining COs generated during transitions from the states $s_m \in S^k$. Finally, the elements of a set $D^k \subseteq D$ include IMFs equal to 1 in the codes of the states next to states $s_m \in S^k$. Each class $S^k \in \pi_S$ determines the following systems of PBFs:

$$D^k = D^k(V^k, B^k); \tag{15}$$

$$Z^k = Z^k(V^k, B^k). \tag{16}$$

To obtain the final values of functions $D_r \in D$ and $z_r \in Z$, it is necessary to create the following SBFs:

$$D = D(D^1, \ldots, D^K); \tag{17}$$

$$Z = Z(Z^1, \ldots, Z^K). \tag{18}$$

The functions $f_i \in D \cup Z$ are disjunctions of corresponding PBFs.

The combined use of these three methods of SD leads to $MP_TY$ Mealy FSMs. The subscript "T" shows that the two-fold state assignment is used. Its structural diagram consists of four logic levels (Figure 3).

In $MP_TY$ Mealy FSM, the block LB generates functions (6) to replace FSM inputs using additional variables. The second logic level consists of blocks LB1, ..., LBK. Each block LBk implements systems of PBFs (15) and (16). These functions are transformed into functions $f_i \in D \cup Z$ by the block LTZ. This block represents the third logic level of FSM circuit. The block LTZ includes two distributed registers. One of them is the state code register RG. The RG outputs are used as a feedback for the input transformation. In addition, they

enter a block LV to create ESCs. The second register (a register RZ) keeps the codes of COs. We discuss the necessity of RZ later. Both registers are zeroed by the pulse *Start* and synchronized by the pulse *Clock*. The fourth logic level includes two blocks. The block LY generates FSM outputs represented by (11). The block LV transforms the maximum state codes $K(s_m)$ into extended state codes $C(s_m)$. This block implements SBF (14).

To reduce the chip area occupied by the LUT-based circuit of $MP_TY$ Mealy FSM, we propose two new approaches. One of them allows the reduction of the number of LUTs and their levels in the circuit of LB. The second method aims to reduce the number of flip-flops necessary for the stabilization of the FSM operation.

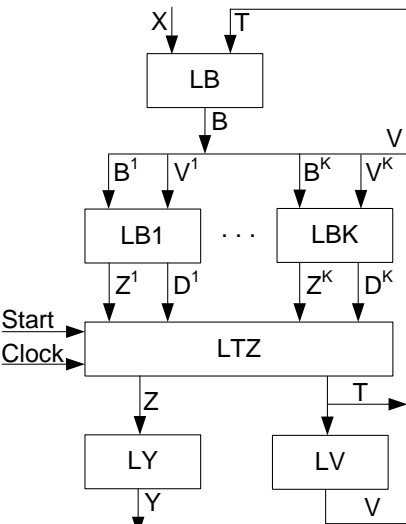

**Figure 3.** Structural diagram of $MP_TY$ Mealy FSM.

We use the symbol $X(b_g)$ for a set of FSM inputs replaced by an additional variable $b_g \in B$. As a rule, the RI is executed in the following way [20]: the number of FSM inputs in different sets $X(b_q)$ should be maximal. At best, identical inputs $x_l \in X$ should be replaced by the same variable $b_g \in B$. Such an approach allows minimization of the chip area if an FSM circuit is implemented using programmable logic arrays (PLAs) [9]. However, PLAs have a lot of inputs, whereas this number is very limited for LUTs. Thus, we propose distributing inputs $x_l \in X$ in a way which allows holding the following condition for the maximum possible number of sets $X(b_g)$:

$$|X(b_g)| + R_{MB} = S_L. \tag{19}$$

Obviously, if (19) takes place for the set $X(b_g)$, then a circuit generating the function $b_g \in B$ includes only one element. If (19) takes place for all sets $X(b_g)$, then the block LB includes $G$ elements. In addition, this circuit is single-level.

To increase the value of $|X(b_g)|$, we propose to encode the states in a way that decreases the number of state variables in functions (6). Let $S(b_g) \subseteq S$ be a set of states whose transitions depend on the inputs $x_l \in X(b_g)$. We propose to encode the states $s_m \in S(b_g)$ in such a way that their codes create the minimum possible number of generalized cubes of $R_{MB}$-dimensional Boolean space. This approach allows excluding some state variables as literals of SOPs (6).

As a rule, FSMs are not stand-alone units. They are used as parts of a digital system. Due to it, the stability of the outputs is one of the very important problems in FSM circuit design [13,42,43]. If an FSM is a part of some digital system, then the FSM outputs are inputs of other system's blocks. It is known [1,20] that outputs of Mealy FSMs are unstable: input fluctuations may lead to output fluctuations. In turn, these fluctuations of FSM outputs may cause failure in some blocks of a digital system. It is possible to avoid such

failures by stabilizing the FSM inputs. To do this, it is necessary to introduce a synchronous register of inputs (RI) [20]. This changes the FSM operation mode.

De facto, the set of inputs $X = \{x_1, \ldots, x_L\}$ consists of outputs of various system blocks. These outputs enter the flip-flops of RI. Till these outputs are transients, the synchronization signal of RI is not active. Due to this, the FSM is disconnected from other blocks. Thus, the RI keeps the values of FSM inputs registered in the previous cycle. After the stabilization of system outputs, they are loaded into the RI using the required edge of synchronization. Thus, eliminating the dependence of the inputs' stability on the stability of system outputs leads to additional area costs and reduces overall performance. This is an overhead of stability (additional LUTs, flip-flops, interconnections, power consumption and delay). Thus, it makes sense to reduce this overhead.

In our paper, we propose to include a register RZ into block LTZ. There is a flip-flop in each CLB generating a function $z_r \in Z$. Thus, to organize the RZ, there is no need for additional LUTs. In addition, these flip-flops could be controlled by already-existing pulses *Start* and *Clock*. Obviously, the proposed approach does not require additional CLBs. This means that it does not require the additional chip area (compared to an FSM architecture which uses either a registration of inputs or a registration of outputs).

A method for the synthesis of $MP_TY$ Mealy FSMs is proposed in this paper. We start the design from an STG [1]. To create tables representing the blocks of the FSM circuit, the STG is transformed into the equivalent STT [1]. The proposed method includes the following steps:

1. Creating STT of Mealy FSM.
2. Executing replacement of FSM inputs.
3. Assignment of maximum binary state codes $K(s_m)$ optimizing SBF (6).
4. Creating SBF (6) representing the block LB.
5. Finding the partition $\pi_S$ with the minimum cardinality number.
6. Assignment of partial codes $C(s_m)$ to states $s_m \in S^k$.
7. Encoding of COs $Y_q \subseteq Y$ using maximum binary codes.
8. Creating SBF (11) representing the block LY.
9. Constructing tables of LB1–LBK and creating SBFs (15) and (16).
10. Constructing the table of LTZ and creating systems (17) and (18).
11. Constructing table of LV and deriving the system (14).
12. Implementing LUT-based circuit of $MP_TY$ FSM.

If an FSM $A$ is synthesized using the model of $MP_TY$ Mealy FSM, then we denote such a situation by the symbol $MP_TY(A)$. Next, we discuss an example of $MP_TY$ FSM synthesis.

## 5. Example of Synthesis of $MP_TY$ Mealy FSM Logic Circuit

We discuss the synthesis of Mealy FSM $MP_TY(A1)$ using LUTs with $S_L = 5$ inputs. The STG (Figure 4) represents the FSM $A1$.

Using STG (Figure 4), we can derive the sets $S = \{s_1, \ldots, s_6\}$ (each vertex of STG corresponds to a state); $X = \{x_1, \ldots, x_8\}$ (these inputs are shown above the STG arcs); and $Y = \{y_1, \ldots, y_9\}$ (these outputs are written above the STG arcs). This gives the following values: $M = 6$, $L = 8$, and $N = 9$. There are $H = 17$ arcs connecting the vertices of STG (Figure 4). Obviously, there are $H = 17$ rows in the equivalent STT. As follows from (1), $R_{MB} = 3$ is necessary to execute the maximum binary state assignment. This gives the sets $T = \{T_1, T_2, T_3\}$ and $D = \{D_1, D_2, D_3\}$.

Step 1. The procedure of transformation is executed using the approach shown in [1]. Each arc of STG determines a row of STT. Each row includes a current state $s_C$, a transition state $s_T$, an input signal $X_h$ which determines the transition from $s_C$ into $s_T$, an output collection $Y_h$, and the row number, $h$. In the discussed example, the STG (Figure 4) is transformed into STT (Table 1). This table includes an additional column $q$ containing the subscripts of COs written in each row of the column $Y_h$.

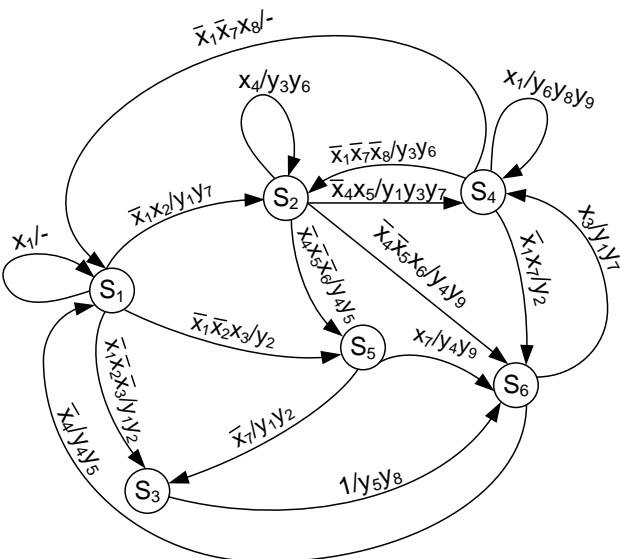

**Figure 4.** State transition graph of Mealy FSM $A1$.

**Table 1.** State transition table of FSM $A1$.

| $S_c$ | $S_T$ | $X_h$ | $Y_h$ | $q$ | $h$ |
|---|---|---|---|---|---|
| $s_1$ | $s_1$ | $x_1$ | - | 1 | 1 |
|  | $s_2$ | $\overline{x_1}x_2$ | $y_1y_7$ | 2 | 2 |
|  | $s_5$ | $\overline{x_1}\,\overline{x_2}x_3$ | $y_2$ | 3 | 3 |
|  | $s_3$ | $\overline{x_1}\,\overline{x_2}\,\overline{x_3}$ | $y_1y_2$ | 4 | 4 |
| $s_2$ | $s_2$ | $x_4$ | $y_3y_6$ | 5 | 5 |
|  | $s_4$ | $\overline{x_4}x_5$ | $y_1y_3y_7$ | 6 | 6 |
|  | $s_6$ | $\overline{x_4}\,\overline{x_5}x_6$ | $y_4y_9$ | 7 | 7 |
|  | $s_5$ | $\overline{x_4}\,\overline{x_5}\,\overline{x_6}$ | $y_4y_5$ | 8 | 8 |
| $s_3$ | $s_6$ | 1 | $y_5y_8$ | 9 | 9 |
| $s_4$ | $s_4$ | $x_1$ | $y_5y_8$ | 10 | 10 |
|  | $s_6$ | $\overline{x_1}x_7$ | $y_2$ | 3 | 11 |
|  | $s_1$ | $\overline{x_1}\,\overline{x_7}x_8$ | - | 1 | 12 |
|  | $s_2$ | $\overline{x_1}\,\overline{x_7}\,\overline{x_8}$ | $y_3y_6$ | 5 | 13 |
| $s_5$ | $s_6$ | $x_7$ | $y_4y_9$ | 7 | 14 |
|  | $s_3$ | $\overline{x_7}$ | $y_1y_2$ | 4 | 15 |
| $s_6$ | $s_4$ | $x_3$ | $y_1y_7$ | 2 | 16 |
|  | $s_1$ | $\overline{x_3}$ | $y_4y_5$ | 8 | 17 |

Step 2. The interstate transitions from $s_m \in S$ depend on inputs creating the set $X(s_m) \subseteq X$ with $NI_m$ elements. To find the number, $G$, of additional variables $b_g \in B$, it is necessary to use the following formula [20]:

$$G = max(NI_1, \ldots, NI_M). \tag{20}$$

As follows from Table 1, the existing sets $X(s_m) \subseteq X$ have the following cardinality numbers: $NI_1 = NI_2 = NI_4 = 3$, $NI_5 = NI_6 = 2$, and $NI_3 = 0$. Using (20) gives $G = 3$ and $B = \{b_1, b_2, b_3\}$.

Thus, there is $S_L = 5$ and $R_{MB} = 3$. Using (19) gives $|X(b_g)| = S_L - R_{MB} = 2$. Thus, the IR should be executed in a way so that the relation $|X(b_g)| = 2$ holds for the maximum possible number of sets $X(b_g)$. Using the proposed approach gives the distribution of inputs shown in Table 2.

**Table 2.** Table of RI for FSM *A*1.

| $B \setminus S$ | $S_1$ | $S_2$ | $S_3$ | $S_4$ | $S_5$ | $S_6$ |
|---|---|---|---|---|---|---|
| $b_1$ | $x_1$ | $x_4$ | - | $x_1$ | - | - |
| $b_2$ | $x_2$ | $x_5$ | - | $x_7$ | $x_7$ | - |
| $b_3$ | $x_3$ | $x_6$ | - | $x_8$ | - | $x_3$ |

Step 3. States $s_m \in S$ should be encoded in a way that minimizes the numbers of literals in SBF (6). We denote by symbol $S(b_g)$ a set of states in which FSM inputs $x_l \in X$ are replaced by the additional variable $b_g \in B$. To optimize SBF (6), we propose placing the codes of states $s_m \in S(b_g)$ in the same rows of an $R_{MB}$- dimensional Karnaugh map. If an input $x_l \in X$ is replaced by a variable $b_g \in B$ for states $s_m, s_i \in S(b_g)$, then we propose placing these states into adjusted cells of the map. To optimize the SOP of $b_g \in B$, we can use three types of insignificant assignments. They are the following: (1) the states with unconditional transitions; (2) the states which do not belong to a particular set $S(b_g)$; and (3) the combinations of state variables which are not used as state codes. For the discussed example, the Karnaugh map (Figure 5) includes the state codes.

**Figure 5.** Outcome of state maximum binary state assignment.

Let us explain how this map was created. There are the sets $S(b_1) = \{s_1, s_2, s_4\}$ and $X(b_1) = \{x_1, x_4\}$. As follows from Figure 5, these states are placed in the same row of the map. For states $s_1$ and $s_4$, the same input $x_1$ is replaced. So, these states have adjacent codes 000 and 010. The code 001 (state $s_3$) can be thought of as insignificant because the transition from this state is unconditional. The code 011 (state $s_5$) can be thought of as insignificant because there is no input symbol in the row $b_1$ ( the transaction from this state is unconditional). To optimize the term depended on $s_2$, we can use state assignments 110 (no state), 111 (the symbol "–" in the row $b_1$) and 101 (no state). As a result, the following Boolean equation is obtained: $b_1 = x_1 \overline{T_1} + x_4 T_1$.

Step 4. Using the approach discussed above, we can obtain the following SBF:

$$
\begin{aligned}
b_1 &= x_1(A_1 \vee A_4) \vee x_4 A_2 = x_1 \overline{T_1} \vee x_4 T_1; \\
b_2 &= x_2 A_1 \vee x_5 A_2 \vee x_7(A_4 \vee A_5) = \\
&= x_2 \overline{T_1}\,\overline{T_2} \vee x_7 T_2; \\
b_3 &= x_3(A_3 \vee A_6) \vee x_2 A_2 \vee x_8 A_4 = \\
&= x_3 \overline{T_1}\,\overline{T_2} T_3 \vee x_3 T_1 T_2 \vee x_2 T_1 \overline{T_2} \vee x_8 x_3 \overline{T_2}\,\overline{T_3}.
\end{aligned}
\tag{21}
$$

The analysis of SBF (21) shows that the circuits implemented into its equations have four LUTs. The circuit for $b_1$ includes a single LUT, as does the circuit for $b_2$. The two-level circuit generating $b_3$ includes two LUTs. Thus, in the discussed case, there are four LUTs and two have their levels in the circuit of LB.

Step 5. We use the approach proposed in the paper [18] to create the partition $\pi_S$. Using the method [18] gives the following sets: $\pi_S = \{S^1, S^2\}$, $S^1 = \{s_1, s_2, s_4\}$ and $S^2 = \{s_3, s_5, s_6\}$. Thus, $K = 2$.

Step 6. As follows from analysis of classes $S^k \in \pi_S$, each class includes $M_k = 3$ states. Using (12) and (13) gives the following: $R_1 = R_2 = 2$, $R_S = 4$, $V^1 = \{v_1, v_2\}$, $V^2 = \{v_3, v_4\}$ and $V = \{v_1, \ldots, v_4\}$. It is known that the partial state codes do not affect

the number of LUTs in the circuits of LBk [18]. Thus, we can assign them in the trivial way: codes are assigned as the subscript grows and corresponds to the decimal number of the step to which the code $C(s_m)$ is assigned. This approach gives the following codes: $C(s_1) = C(s_3) = 01, C(s_2) = C(s_5) = 10$, and $C(s_4) = C(s_6) = 11$.

Step 7. As follows from Table 1, during the operation of the FSM $A1$, the following COs are generated: $Y_1 = \{\}$, $Y_2 = \{y_1, y_7\}$, $Y_3 = \{y_2\}$, $Y_4 = \{y_1, y_2\}$, $Y_5 = \{y_3, y_6\}$, $Y_6 = \{y_1, y_3, y_7\}$, $Y_7 = \{y_4, y_9\}$, $Y_8 = \{y_4, y_5\}$, $Y_9 = \{y_5, y_8\}$, $Y_{10} = \{y_6, y_8, y_9\}$. Thus, there are $Q = 10$ collections of outputs generated during the interstate transitions of FSM $A1$. Using (9) gives $R_{CO} = 4$ and the set $Z = \{z_1, \ldots, z_4\}$.

The encoding is executed in such a way as to reduce the total number of literals in SOPs (11). This can be carried out using, for example, the approach from the work [44]. One of the possible outcomes is shown in (Figure 6).

**Figure 6.** Codes of output collections.

Step 8. Using codes $K(Y_q)$ and insignificant input assignments [1], we can obtain the following SBF:

$$\begin{aligned}
y_1 &= Y_2 \vee Y_4 Y_6 = \overline{z_1} z_2; \\
y_2 &= Y_3 \vee Y_4 = \overline{z_1} z_4; \\
y_3 &= Y_5 \vee Y_6 = \overline{z_1} z_3; \\
y_4 &= Y_7 \vee Y_8 = z_1 \overline{z_3}; \\
y_5 &= Y_8 \vee Y_9 = z_1 z_4; \\
y_6 &= Y_5 \vee Y_{10} = \overline{z_2} z_3; \\
y_7 &= Y_2 \vee Y_6 = z_2 \overline{z_4}; \\
y_8 &= Y_9 \vee Y_{10} = z_1 z_3; \\
y_9 &= Y_7 \vee Y_{10} = z_1 \overline{z_4}.
\end{aligned} \qquad (22)$$

The SBF (22) represents the circuit of block LY. Thus, it corresponds to SBF (11). The maximum number of literals in the SOPs of (11) is determined as $N \times R_{CO}$. In the discussed case, this number is equal to $9 \times 4 = 36$. The SBF (22) contains 18 literals. Thus, using the approach [44] allows a reduction in the number of literals by a factor of 2.0 compared to its maximum possible value. Each literal corresponds to the interconnection between the blocks LTZ and LY. Thus, reducing the number of literals results in reducing the number of interconnections. This is a positive factor because interconnections significantly influence the chip area used, power consumption and performance.

Step 9. To create a table of LBk, it is necessary to use the STT rows representing transitions from states $s_m \in S^k$. For example, to create a table representing LB1, we should choose the rows 1–8 and 10–13 of Table 1. The column $X_h$ should be replaced by the column $B_h^1$. This column includes the conjunctions of variables $b_g \in B$ corresponding the conjunctions of replaced inputs $x_l \in X$. The column $Y_h$ is replaced by the column $Z_h^1$. This column includes the variables $z_r \in Z$ equal to 1 in the codes $K(Y_q)$ of COs shown the corresponding rows of STT.

In addition, this table includes the columns $C(s_C)$ (the partial code of the current state), $K(s_T)$ (the MBC of the next state), and $D^1_h$ (IMFs equal to 1 to load the code $K(s_T)$ into RG). In the discussed case, this table contains H1 = 12 rows (Table 3).

For example, the second row of Table 3 is created in the following manner. This row is constructed using the second row of Table 1. This row describes the transition $\langle s_1, s_2 \rangle$ executed when the following relation takes place: $\overline{x_1} x_2 = 1$. During this transition, the CO $Y_2 = \{y_4, y_4\}$ is produced. From the outcome of step 6, we have the code $C(s_1) = 01$. This code should be placed in the column $C(s_C)$. Using the Karnaugh map (Figure 5) gives state code $K(s_T) = 100$. This code should be placed in the column $K(s_T)$. It determines existence of the symbol $D_1$ in the column $D^1_h (h = 2)$ of Table 3. As follows from the column $s_1$ of Table 2, the input $x_1$ is represented by $b_1$ and the input $x_2$ is replaced by the variable $b_2$. Thus, the conjunction $\overline{x_1} x_2$ is replaced by the conjunction $\overline{b_1} b_2$ written in the column $B^1_h (h = 1)$ of Table 3.

**Table 3.** Table of block LB1.

| $S_c$ | $C(S_c)$ | $S_T$ | $K(S_T)$ | $B^1_h$ | $Z^1_h$ | $D^1_h$ | $h$ |
|---|---|---|---|---|---|---|---|
| $s_1$ | 01 | $s_1$ | 000 | $b_1$ | - | - | 1 |
| | | $s_2$ | 100 | $\overline{b_1} b_2$ | $z_2$ | $D_1$ | 2 |
| | | $s_5$ | 011 | $\overline{b_1}\,\overline{b_2} b_3$ | $z_4$ | $D_2 D_3$ | 3 |
| | | $s_3$ | 001 | $\overline{b_1}\,\overline{b_2}\,\overline{b_3}$ | $z_2 z_4$ | $D_3$ | 4 |
| $s_2$ | 10 | $s_2$ | 100 | $b_1$ | $z_3$ | $D_1$ | 5 |
| | | $s_4$ | 010 | $\overline{b_1} b_2$ | $z_2 z_3$ | $D_2$ | 6 |
| | | $s_6$ | 111 | $\overline{b_1}\,\overline{b_2} b_3$ | $z_1$ | $D_1 D_2 D_3$ | 7 |
| | | $s_5$ | 011 | $\overline{b_1}\,\overline{b_2}\,\overline{b_3}$ | $z_1 z_4$ | $D_2 D_3$ | 8 |
| $s_4$ | 11 | $s_4$ | 010 | $b_1$ | $z_1 z_3$ | $D_2$ | 9 |
| | | $s_6$ | 111 | $\overline{b_1} b_2$ | $z_4$ | $D_1 D_2 D_3$ | 10 |
| | | $s_1$ | 000 | $\overline{b_1}\,\overline{b_2} b_3$ | - | - | 11 |
| | | $s_2$ | 100 | $\overline{b_1}\,\overline{b_2}\,\overline{b_3}$ | $z_1$ | $D_1$ | 12 |

A similar approach is used to create all the rows of Table 3 (block LB1) and Table 4 (block LB2). These tables represent SBFs (15) and (16). There are examples of some SOPs shown below:

$$
\begin{aligned}
z^1_1 &= v_1 \overline{v_2}\overline{b_1}\,\overline{b_2} \vee v_1 v_2 b_1 \vee v_1 v_2 \overline{b_2}\,\overline{b_3}; \\
D^1_3 &= \overline{v_1} v_2 \overline{b_1}\,\overline{b_2} \vee v_1 \overline{v_2}\overline{b_1}\,\overline{b_2} \vee v_1 v_2 \overline{b_1} b_2.
\end{aligned} \tag{23}
$$

$$
\begin{aligned}
z^2_1 &= \overline{v_3} v_4 \vee v_3 \overline{v_4} b_2 \vee v_3 v_4 \overline{b_3}; \\
D^2_3 &= \overline{v_3} v_4 \vee v_3 \overline{v_4}.
\end{aligned} \tag{24}
$$

Step 10. The table of block LTZ includes the following columns: "Function" (the column includes symbols $D_r \in D$ and $z_r \in Z$), LB1, LB2. If a PBF is generated by the block LBk ($k \in \{0, 1, \ldots, K\}$), then the intersection of the row with this function and the column LBk is marked by 1. Otherwise, this intersection contains zero. The block LTZ is represented by Table 5.

**Table 4.** Table of block LB2.

| $S_c$ | $C(S_c)$ | $S_T$ | $K(S_T)$ | $B_h^2$ | $Z_h^2$ | $D_h^2$ | $h$ |
|-------|----------|-------|----------|---------|---------|---------|-----|
| $s_3$ | 01 | $s_6$ | 111 | 1 | $z_1 z_2 z_3 z_4$ | $D_1 D_2 D_3$ | 1 |
| $s_5$ | 10 | $s_6$ | 111 | $b_1$ | $z_3$ | $D_1 D_2 D_3$ | 2 |
|       |    | $s_3$ | 001 | $\overline{b_2}$ | $z_2 z_4$ | $D_3$ | 3 |
| $s_6$ | 11 | $s_4$ | 010 | $b_3$ | $z_2 z_4$ | $D_2$ | 4 |
|       |    | $s_6$ | 000 | $\overline{b_3}$ | $z_1 z_7$ | - | 5 |

To fill the columns LB1 and LB2, we use Tables 3 and 4, respectively. In the discussed case, Table 5 determines SBFs (17) and (18). For example, the following disjunctions may be derived from Table 5:

$$
\begin{aligned}
z_1 &= z_1^1 \vee z_1^2; \\
D_3 &= D_3^1 \vee D_3^2.
\end{aligned}
\tag{25}
$$

Step 11. The block LV converts MBC codes $K(s_m)$ into the partial state codes $C(s_m)$. The conversion is executed for all states. The table of LV includes the columns $s_m$, $K(s_m)$, $C(s_m)$, $V_m$. If there is $v_r = 1$ for a particular code $C(s_m)$, then there is the symbol $v_r$ in the column $V_m$ (Table 6).

**Table 5.** Table of LTZ.

| Function | LB1 | LB2 |
|----------|-----|-----|
| $D_1$ | 1 | 1 |
| $D_2$ | 1 | 1 |
| $D_3$ | 1 | 1 |
| $z_1$ | 1 | 1 |
| $z_2$ | 1 | 1 |
| $z_3$ | 1 | 1 |
| $z_4$ | 1 | 1 |

**Table 6.** Table of block LV.

| $S_m$ | $K(S_m)$ | $C(S_m)$ | $V_m$ |
|-------|----------|----------|-------|
| $s_1$ | 000 | 0100 | $v_2$ |
| $s_2$ | 100 | 1000 | $v_1$ |
| $s_3$ | 001 | 0001 | $v_4$ |
| $s_4$ | 010 | 1100 | $v_1 v_2$ |
| $s_5$ | 011 | 0010 | $v_3$ |
| $s_6$ | 111 | 0011 | $v_3 v_4$ |

Using Table 6, it is possible to create SBF (14) represented by its perfect SOPs. To minimize these SOPs, we can create a multi-functional Karnaugh map, as shown in Figure 7.

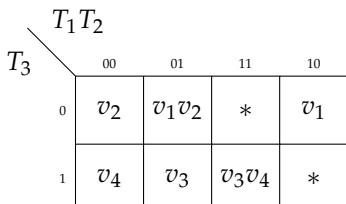

**Figure 7.** Multi-functional map of LV.

This Karnaugh map is created using the codes from Figure 5. In Figure 7, the symbols of states $s_m \in S$ are replaced by symbols of additional variables $v_r \in V$. This is performed in the following way: if a particular cell of Figure 5 includes a state $s_m \in S^k$, then the symbols $v_r \in V^k$ are rewritten into the corresponding cell of Figure 7. Using Figure 7 gives the following SBF, which determines the contents of LUTs from the block LV:

$$\begin{aligned}
v_1 &= A_2 \vee A_4 = T_1\overline{T_3} \vee T_2\overline{T_3}; \\
v_2 &= A_1 \vee A_4 = \overline{T_1} \vee \overline{T_3}; \\
v_3 &= A_5 \vee A_6 = T_2 T_3; \\
v_4 &= A_3 \vee A_6 = \overline{T_2} T_3 \vee T_1 T_3.
\end{aligned} \qquad (26)$$

Step 12. Using the obtained SOPs, we can estimate how many LUTs it is necessary to implement in the circuit of $MP_TY(A1)$. As follows from SBF (21), condition (19) holds for SOP functions $b_1, b_2 \in B$. Thus, each of these functions is implemented using a single LUT with $S_L = 5$. There are six literals in the SOP $b_3 \in B$. Thus, this SOP should be decomposed. As a result, the corresponding circuit includes two LUTs connected in series. Due to this, the circuit of LB includes four LUTs and has two levels of logic (Figure 8).

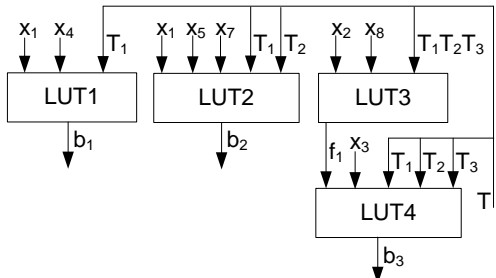

**Figure 8.** Circuit of block LB for Mealy FSM $MP_TY(A1)$.

Each of the blocks LB1, LB2 (the second level of logic) and LTZ (the third level of logic) have circuits with seven LUTs. Each of these circuits is single-level. The fourth level consists of circuits for blocks LY (nine LUTs) and LV (four LUTs).

Thus, the resulting circuit has five levels and includes 38 LUTs. Our analysis of Mealy FSM MPY(A1) shows the following. There are the same LUT counts for the circuits of the blocks LB and LY of equivalent MPY and $MP_TY$ FSMs. Thus, in the discussed case, these blocks include $4 + 9 = 13$ LUTs. There are $R_{MB} + G = 6$ literals in the SOPs of SBFs (7) and (10). Using LUTs with five inputs leads to the functional decomposition of these SOPs. As the result, there are three LUTs in a two-level circuit implementing any function from SBFs (7) and (10). There are $R_{MB} + R_{CO} = 7$ functions generated by the LTZ of Mealy FSM MPY(A1). Thus, there are 21 LUTs in this circuit. This calculation gives 34 LUTs in the circuit of Mealy FSM MPY(A1). The circuit has five levels of LUTs.

Thus, there is the same number of levels in the circuits of FSMs MPY(A1) and $MP_TY(A1)$. However, the circuit of Mealy FSM MPY(A1) includes fewer LUTs. It is possible to obtain the same LUT count for both circuits if we change the approach for the encoding of states and COs [16]. However, we do not discuss this approach in our current paper.

Our example is rather simple. It is necessary to compare equivalent FSMs based on various approaches using some benchmarks with a wide range of characteristics. Such a comparison is given in the next Section. This comparison is executed for FPGAs produced by AMD Xilinx. Due to this, the industrial package Vivado [39] is applied to fulfil all the necessary steps of technology mapping [7,26,45].

## 6. Experimental Results

To compare the LUT-based circuits produced by our proposed method with circuits obtained using some known design methods, we use 48 benchmarks creating the library LGSsynth93 [31]. These benchmarks have a wide diapason of their main characteristics such as: the numbers of transitions, internal states, input variables, output functions, collections of FSM outputs. The benchmarks are represented by STTs in the format KISS2. The choice of this library is based on the fact that a lot of FSM designers use it to compare their results with main characteristics of known FSM circuits [27,36,37,46–48]. The characteristics of the benchmark FSMs could be found, for example, in our previous articles. Due to this, we do not show them in our current paper.

To conduct the experiments, we use the Virtex-7 VC709 platform (xc7vx690tffg1761-2) [49] based on FPGA chip xc7vx690tffg1761-2 (AMD Xilinx). The CLBs of this chip include LUTs with six address inputs. To obtain the FSM circuits, we use an industrial package Vivado v2019.1 (64-bit) [39] produced by AMD Xilinx. To process the benchmarks, we use their VHDL-based models. To transform the KISS2-based benchmarks files into VHDL codes, the CAD tool K2F [50] is applied.

For each benchmark, we use Vivado reports to find the LUT counts and performance (the values of cycle time and maximum operating frequency). We compare the proposed FSM model with four different FSM models. Three of these models are P FSMs based on: (1) Auto of Vivado (P Mealy FSMs with MBCs); (2) One-hot of Vivado (one-hot-based P Mealy FSMs); (3) JEDI (P Mealy FSMs with MBCs). As the fourth model, we investigate the MPY Mealy FSMs.

In our research, we take into account the fact that FSMs are not stand-alone units. To achieve the stability of the outputs, we use an additional synchronous register. In the cases of P FSMS, the inputs are loaded into this register. Thus, it consists of L flip-flops. Obviously, to implement this register, it is necessary to use L additional LUTs. In the cases of both MPY and $MP_TY$ FSMs, this register keeps the codes of COs. Thus, it has $R_{CO}$ flip-flops and does not require additional LUTs. In addition, it does not require the additional synchronization pulse. This simplifies the synchronization circuit compared with equivalent P FSMs.

The results of experiments [16,17] show that practically all the characteristics of LUT-based FSM circuits strongly depend on the relation between the values of $L + R_{MB}$, on the one hand, and $S_L$, on the other hand. In experiments, we use Virtex-7 FPGAs for which $S_L = 6$. We divided the set of benchmarks by classes of complexity (CC). If the symbol CCP ($P = 1, 2, \ldots$) means a class number, then the benchmarks belonging to a certain class is determined by the expression

$$CCP = \lceil (L + R_{MB})/S_L \rceil - 1. \tag{27}$$

For the library used, there are five classes of complexity (CC0-CC4). In each of the following tables, the benchmarks belonging to a certain class are shown in the column "Class of complexity". The class CC0 includes trivial FSMs. The class CC1 includes simple FSMs. The class CC2 includes average FSMs. The class CC3 includes big FSMs. Finally, the class CC4 includes very big FSMs.

Tables 7–16 contain the results of the experiments conducted. Table 7 includes the numbers of LUTs necessary to implement the electrical circuit for a given benchmark. All benchmarks are represented in this table. Table 8 contains the LUT counts for classes CC0–CC1. Table 9 contains the LUT counts for classes CC2–CC4. The negative influence of the number of FSM inputs is shown in Table 10. Table 11 contains the values of the minimum cycle times for each benchmark. The data for these tables are taken from the Vivado reports. In addition, we show cycle times separately for classes CC0–CC1 (Table 12) and CC2–CC4 (Table 13). The values of the maximum operating frequencies are shown in Table 14. These values are obtained in a simple way using data from Table 11. In addition, we show the frequencies separately for classes CC0–CC1 (Table 15) and CC2–CC4 (Table 16).

Each table is organized in the same manner. The first column includes the benchmarks' names, the row "Total" and the row "Percentage". The names of the investigated methods are shown in the next five columns. The classes of complexity are shown in the last column. In the row "Total" are shown the results of the summation of values for a particular column. Finally, the row "Percentage" includes the percentage of the summarized characteristics of various FSM circuits in relation to the summarized characteristics of $MP_T Y$ FSMs. We start the discussion of the results starting with Table 7.

As follows from Table 8, as compared to other investigated methods, the circuits of $MP_T Y$-based FSMs consist of the minimum number of LUTs. There is the following gain: (1) 56.99% compared to Auto-based FSMs; (2) 79.13% compared to One-hot –based FSMs; (3) 33.13% compared to JEDI-based FSMs; and (4) 8.98% compared to MPY-based FSMs. In second place in terms of gain are MPY-based FSMs. We think this gain is associated with two factors. First, for rather complex FSMs, SD-based circuits always have fewer LUTs than for equivalent FD-based FSMs [9]. Second, there are an additional L LUTs in the circuits of FD-based FSMs required to stabilize their operation. In the case of both MPY- and $MP_T Y$-based FSMs, the stabilization is achieved by registering the codes of COs. To produce these codes, LUTs of LTZ are used. The outputs of these LUTs are connected with $R_{CO}$ flip-flops creating the additional register. Thus, there is no need for additional LUTs. Of course, the gain is also associated with replacing FSM inputs with additional variables. We think that this diminishes the number of partial functions compared to equivalent FD-based FSMs.

It is interesting to show how the gain is changed with the change in FSM complexity. Using Table 7, we created two additional tables. Table 9 shows LUT counts for trivial and simple FSMs. Table 9 contains information about LUT counts for average, big and very big FSMs.

Analysis of Table 8 shows that the proposed approach provides the same LUT counts as for equivalent MPY FSMs. All P-based models require more LUTs. Our approach gives the following gain: (1) 24.89% compared to Auto-based FSMs; (2) 56.11% compared to One-hot—based FSMs; and (3) 9.61% compared to JEDI-based FSMs. We think that this gain is connected to the different stabilization methods used in SD- and FD-based FSMs. The input register of FD-based FSMs requires more LUTs than the output register of SD-based FSMs. However, both MPY- and $MP_T Y$-based FSMs require more LUTs for trivial FSMs (the complexity class CC0). We think this has a very simple explanation. Namely, for trivial FSMs, the condition (5) holds. Thus, there is no need to apply the SD-based methods. However, these methods are always used during the synthesis of both MPY- and $MP_T Y$-based FSMs. In this case, it is necessary to implement circuits of blocks LB and LY. It is the presence of these absolutely redundant blocks that determines the marked loss of SD-based methods.

The next phenomenon comes from Table 8: for the class CC0, the circuits of equivalent MPY- and $MP_T Y$-based FSMs have equal amounts of LUTs. We think this is connected with the fact that the partition $\pi_S$ consists of one class. Due to this, there is no need to use the blocks LB1–LBK. This means that $MP_T Y$ FSMs turn into MPY FSMs. Obviously, these FSM circuits should have equal values for all the other characteristics. This, once again, indicates that it is advisable to use different FSM models for different conditions. Thus, it makes no sense to apply SD-based methods when condition (5) is met.

**Table 7.** Experimental results (the LUT counts).

| Benchmark | Auto | One-Hot | JEDI | MPY | MP$_T$Y | Class of Complexity |
|-----------|------|---------|------|-----|---------|---------------------|
| bbara | 21 | 21 | 14 | 12 | 12 | CC1 |
| bbsse | 40 | 44 | 31 | 14 | 14 | CC1 |
| bbtas | 7 | 7 | 7 | 9 | 9 | CC0 |
| beecount | 22 | 22 | 17 | 13 | 13 | CC1 |
| cse | 47 | 73 | 43 | 18 | 18 | CC1 |
| dk14 | 19 | 30 | 13 | 12 | 12 | CC1 |
| dk15 | 18 | 19 | 15 | 11 | 11 | CC1 |
| dk16 | 17 | 36 | 14 | 14 | 14 | CC1 |
| dk17 | 7 | 14 | 7 | 9 | 9 | CC0 |
| dk27 | 4 | 6 | 5 | 8 | 8 | CC0 |
| dk512 | 11 | 11 | 10 | 14 | 14 | CC0 |
| donfile | 33 | 33 | 26 | 21 | 21 | CC1 |
| ex1 | 79 | 83 | 62 | 28 | 24 | CC2 |
| ex2 | 11 | 11 | 10 | 11 | 11 | CC1 |
| ex3 | 11 | 11 | 11 | 16 | 16 | CC0 |
| ex4 | 21 | 19 | 18 | 12 | 12 | CC1 |
| ex5 | 11 | 11 | 11 | 15 | 15 | CC0 |
| ex6 | 29 | 41 | 27 | 21 | 21 | CC1 |
| ex7 | 6 | 7 | 6 | 10 | 10 | CC1 |
| keyb | 50 | 68 | 47 | 28 | 28 | CC1 |
| kirkman | 54 | 70 | 51 | 28 | 22 | CC2 |
| lion | 4 | 7 | 4 | 10 | 10 | CC0 |
| lion9 | 8 | 13 | 7 | 12 | 12 | CC0 |
| mark1 | 28 | 28 | 25 | 22 | 22 | CC1 |
| mc | 7 | 10 | 7 | 12 | 12 | CC0 |
| modulo12 | 8 | 8 | 8 | 11 | 11 | CC0 |
| opus | 33 | 33 | 27 | 20 | 20 | CC1 |
| planet | 138 | 138 | 95 | 76 | 68 | CC2 |
| planet1 | 138 | 138 | 95 | 76 | 68 | CC2 |
| pma | 102 | 102 | 94 | 74 | 62 | CC2 |
| s1 | 73 | 107 | 69 | 52 | 48 | CC2 |
| s1488 | 132 | 139 | 116 | 86 | 79 | CC2 |
| s1494 | 134 | 140 | 118 | 92 | 83 | CC2 |
| s1a | 57 | 89 | 51 | 42 | 35 | CC2 |
| s208 | 23 | 42 | 21 | 20 | 18 | CC2 |
| s27 | 10 | 22 | 10 | 12 | 12 | CC1 |
| s386 | 33 | 46 | 29 | 31 | 31 | CC1 |
| s420 | 29 | 50 | 28 | 24 | 20 | CC4 |

**Table 7.** *Cont.*

| Benchmark | Auto | One-Hot | JEDI | MPY | MP$_T$Y | Class of Complexity |
|:---:|:---:|:---:|:---:|:---:|:---:|:---:|
| s510 | 67 | 67 | 51 | 42 | 36 | CC4 |
| s820 | 13 | 13 | 13 | 14 | 14 | CC1 |
| s832 | 106 | 100 | 86 | 70 | 62 | CC4 |
| s840 | 98 | 97 | 80 | 68 | 56 | CC4 |
| sand | 143 | 143 | 125 | 99 | 83 | CC3 |
| shiftreg | 3 | 7 | 3 | 8 | 8 | CC0 |
| sse | 40 | 44 | 37 | 38 | 38 | CC1 |
| styr | 102 | 129 | 90 | 81 | 79 | CC2 |
| tma | 52 | 46 | 46 | 41 | 36 | CC2 |
| Total | 2099 | 2395 | 1780 | 1457 | 1337 | |
| Percentage, % | 156.99 | 179.13 | 133.13 | 108.98 | 100.00 | |

Now, we are going to discuss the temporal characteristics of FSM circuits. First of all, we show the negative influence of input register. In all P-based FSMs, the stabilization of operation is achieved due to loading FSM inputs into the additional register. Thus, this approach leads to the use of L additional LUTs and flip-flops. Obviously, the cycle time increases due to the presence of the chain < input-LUTs–flip-flops–LUTs of LB>. In addition, this increases the consumed power. We explored how the number of inputs affects the time and power characteristics of resulting circuits. This information is shown in Table 10.

As follows from Table 10, the number of inputs significantly affects the timing and energy characteristics of LUT-based FSM circuits. The more inputs the FSM has, the greater their negative impact. In the case of the investigated SD-based FSMs, the stabilization is achieved due to the registering codes of COs. In this case, the number of additional flip-flops is equal to $R_{CO}$. Moreover, there is no need for additional LUTs because the codes of COs are generated by the LUTs of LTZ. As follows, for the studied benchmarks, the following relation holds: $R_{CO} \ll L$. The validity of this relation determines the gain in time characteristics obtained due to the transition from FD-based FSMs to SD-based FSMs. This gain is shown in Table 11.

As follows from Table 11, the SD-based FSMs have the best values of cycle time. Our proposed method produces FSM circuits which are a bit slower than the circuits of MPY-based FSMs (the average loss is 0.76%). However, our method has the following average gain compared to other FSMs: (1) 70.65% compared to Auto-based FSMs; (2) 71.08% compared to One-hot-based FSMs; and (3) 62.13% compared to JEDI-based FSMs. This gain for the SD-based FSMs is explained by the difference in the methods used for stabilizing the FSM outputs, as discussed before.

To show the influence of FSM complexity, we create two additional tables. Table 12 includes information about the cycle times for trivial and simple FSMs. Table 13 includes information about the cycle times for average, big and very big FSMs.

As follows from Table 12, the time characteristics are equal for SD-based trivial and simple FSMs. They have the following gain: (1) 65.63% compared with both Auto- and One-hot—based FSMs and (2) 59.60% compared with JEDI-based FSMs. The reasons for this situation are as discussed before.

**Table 8.** Experimental results (the LUT counts for classes CC0-CC1).

| Benchmark | Auto | One-Hot | JEDI | MPY | MP$_T$Y | Class of Complexity |
|---|---|---|---|---|---|---|
| bbara | 21 | 21 | 14 | 12 | 12 | CC1 |
| bbsse | 40 | 44 | 31 | 14 | 14 | CC1 |
| bbtas | 7 | 7 | 7 | 9 | 9 | CC0 |
| beecount | 22 | 22 | 17 | 13 | 13 | CC1 |
| cse | 47 | 73 | 43 | 18 | 18 | CC1 |
| dk14 | 19 | 30 | 13 | 12 | 12 | CC1 |
| dk15 | 18 | 19 | 15 | 11 | 11 | CC1 |
| dk16 | 17 | 36 | 14 | 14 | 14 | CC1 |
| dk17 | 7 | 14 | 7 | 9 | 9 | CC0 |
| dk27 | 4 | 6 | 5 | 8 | 8 | CC0 |
| dk512 | 11 | 11 | 10 | 14 | 14 | CC0 |
| donfile | 33 | 33 | 26 | 21 | 21 | CC1 |
| ex2 | 11 | 11 | 10 | 11 | 11 | CC1 |
| ex3 | 11 | 11 | 11 | 16 | 16 | CC0 |
| ex4 | 21 | 19 | 18 | 12 | 12 | CC1 |
| ex5 | 11 | 11 | 11 | 15 | 15 | CC0 |
| ex6 | 29 | 41 | 27 | 21 | 21 | CC1 |
| ex7 | 6 | 7 | 6 | 10 | 10 | CC1 |
| keyb | 50 | 68 | 47 | 28 | 28 | CC1 |
| lion | 4 | 7 | 4 | 10 | 10 | CC0 |
| lion9 | 8 | 13 | 7 | 12 | 12 | CC0 |
| mark1 | 28 | 28 | 25 | 22 | 22 | CC1 |
| mc | 7 | 10 | 7 | 12 | 12 | CC0 |
| modulo12 | 8 | 8 | 8 | 11 | 11 | CC0 |
| opus | 33 | 33 | 27 | 20 | 20 | CC1 |
| s27 | 10 | 22 | 10 | 12 | 12 | CC1 |
| s386 | 33 | 46 | 29 | 31 | 31 | CC1 |
| s820 | 13 | 13 | 13 | 14 | 14 | CC1 |
| shiftreg | 3 | 7 | 3 | 8 | 8 | CC0 |
| sse | 40 | 44 | 37 | 38 | 38 | CC1 |
| Total | 572 | 715 | 502 | 458 | 458 | |
| Percentage, % | 124.89 | 156.11 | 109.61 | 100.00 | 100.00 | |

As follows from Table 13, starting from the complexity CC2, our approach wins in performance. There is the following gain: (1) 78.93% compared with Auto-based FSMs; (2) 79.72% compared with One-hot-based FSMs; (3) 66.3% compared with JEDI-based FSMs and (4) 2.0% compared with equivalent MPY FSMs. We think that the superiority of SD-based FSMs is due to the fact that they generate fewer partial Boolean functions. Due to this, their circuits have fewer logic levels and interconnections. In turn, they are faster.

The slight superiority of MP$_T$Y FSMs (2%) in relation to MPY FSMs is due to the fact that MP$_T$Y FSMs have fewer interconnections. This is connected with different approaches of stabilization. Since interconnections significantly affect the timing characteristics, our

approach produces faster circuits for FSMs from the classes CC2-CC4. Apparently, equivalent SD-based FSMs have the same number of logic levels (the number of series-connected LUTs). Thus, with respect to the other methods under study, the performance of $MP_TY$ FSMs improves as their complexity increases.

**Table 9.** Experimental results (the LUT counts for classes CC2-CC4).

| Benchmark | Auto | One-Hot | JEDI | MPY | $MP_TY$ | Class of Complexity |
|---|---|---|---|---|---|---|
| ex1 | 79 | 83 | 62 | 28 | 24 | CC2 |
| kirkman | 54 | 70 | 51 | 28 | 22 | CC2 |
| planet | 138 | 138 | 95 | 76 | 68 | CC2 |
| planet1 | 138 | 138 | 95 | 76 | 68 | CC2 |
| pma | 102 | 102 | 94 | 74 | 62 | CC2 |
| s1 | 73 | 107 | 69 | 52 | 48 | CC2 |
| s1488 | 132 | 139 | 116 | 86 | 79 | CC2 |
| s1494 | 134 | 140 | 118 | 92 | 83 | CC2 |
| s1a | 57 | 89 | 51 | 42 | 35 | CC2 |
| s208 | 23 | 42 | 21 | 20 | 18 | CC2 |
| s420 | 29 | 50 | 28 | 24 | 20 | CC4 |
| s510 | 67 | 67 | 51 | 42 | 36 | CC4 |
| s832 | 106 | 100 | 86 | 70 | 62 | CC4 |
| s840 | 98 | 97 | 80 | 68 | 56 | CC4 |
| sand | 143 | 143 | 125 | 99 | 83 | CC3 |
| styr | 102 | 129 | 90 | 81 | 79 | CC2 |
| tma | 52 | 46 | 46 | 41 | 36 | CC2 |
| Total | 1527 | 1680 | 1278 | 999 | 879 | |
| Percentage, % | 173.72 | 191.13 | 145.39 | 113.65 | 100.00 | |

**Table 10.** Influence of input register on cycle time and consumed power.

| L | Power [W] | Data Path Delay [ns] |
|---|---|---|
| 1 | 0.356 | 3.471 |
| 2 | 0.367 | 3.599 |
| 3 | 0.380 | 3.603 |
| 4 | 0.392 | 3.640 |
| 5 | 0.406 | 3.667 |
| 6 | 0.418 | 3.688 |
| 7 | 0.431 | 3.729 |
| 8 | 0.448 | 3.793 |
| 9 | 0.462 | 3.800 |
| 10 | 0.477 | 3.705 |
| 11 | 0.491 | 3.767 |
| 12 | 0.511 | 3.898 |
| 18 | 0.608 | 4.112 |
| 19 | 0.623 | 4.113 |

**Table 11.** Experimental results (the cycle time, nanoseconds).

| Benchmark | Auto | One-Hot | JEDI | MPY | MP$_T$Y | Class of Complexity |
|---|---|---|---|---|---|---|
| bbara | 8.811 | 8.811 | 8.352 | 5.214 | 5.214 | CC1 |
| bbsse | 10.096 | 9.642 | 9.213 | 5.226 | 5.226 | CC1 |
| bbtas | 8.497 | 8.497 | 8.451 | 5.308 | 5.308 | CC0 |
| beecount | 9.605 | 9.605 | 8.941 | 5.373 | 5.373 | CC1 |
| cse | 10.558 | 9.840 | 9.343 | 5.453 | 5.453 | CC1 |
| dk14 | 8.821 | 9.395 | 8.762 | 5.839 | 5.839 | CC1 |
| dk15 | 8.797 | 8.998 | 8.735 | 5.219 | 5.219 | CC1 |
| dk16 | 9.491 | 9.320 | 8.672 | 5.245 | 5.245 | CC1 |
| dk17 | 8.617 | 9.587 | 8.617 | 5.400 | 5.400 | CC0 |
| dk27 | 8.325 | 8.424 | 8.369 | 5.195 | 5.195 | CC0 |
| dk512 | 8.566 | 8.566 | 8.477 | 4.119 | 4.119 | CC0 |
| donfile | 9.033 | 9.034 | 8.509 | 5.168 | 5.168 | CC1 |
| ex1 | 10.425 | 10.955 | 9.454 | 5.821 | 5.741 | CC2 |
| ex2 | 8.635 | 8.635 | 8.596 | 5.624 | 5.624 | CC1 |
| ex3 | 8.731 | 8.731 | 8.707 | 5.931 | 5.931 | CC0 |
| ex4 | 9.214 | 9.315 | 8.874 | 5.481 | 5.481 | CC1 |
| ex5 | 9.147 | 9.147 | 9.119 | 5.425 | 5.425 | CC0 |
| ex6 | 9.564 | 9.772 | 9.330 | 5.369 | 5.369 | CC1 |
| ex7 | 8.598 | 8.578 | 8.584 | 5.200 | 5.200 | CC1 |
| keyb | 10.121 | 10.699 | 9.666 | 5.265 | 5.265 | CC1 |
| kirkman | 10.971 | 10.392 | 10.280 | 5.612 | 5.482 | CC2 |
| lion | 8.539 | 8.501 | 8.541 | 6.062 | 6.062 | CC0 |
| lion9 | 8.470 | 8.998 | 8.444 | 5.270 | 5.270 | CC0 |
| mark1 | 9.825 | 9.825 | 9.343 | 6.395 | 6.395 | CC1 |
| mc | 8.688 | 8.719 | 8.682 | 6.099 | 6.099 | CC0 |
| modulo12 | 8.302 | 8.302 | 8.299 | 5.928 | 5.928 | CC0 |
| opus | 9.684 | 9.684 | 9.275 | 5.322 | 5.322 | CC1 |
| planet | 11.264 | 11.264 | 9.073 | 6.018 | 5.878 | CC2 |
| planet1 | 11.264 | 11.264 | 9.073 | 6.018 | 5.834 | CC2 |
| pma | 10.634 | 10.634 | 9.681 | 6.101 | 6.101 | CC2 |
| s1 | 10.623 | 11.154 | 10.156 | 5.830 | 5.707 | CC2 |
| s1488 | 11.013 | 11.372 | 10.155 | 6.432 | 6.206 | CC2 |
| s1494 | 10.487 | 10.654 | 9.878 | 5.723 | 5.511 | CC2 |
| s1a | 10.313 | 9.462 | 9.704 | 5.689 | 5.511 | CC2 |
| s208 | 9.503 | 9.434 | 9.361 | 6.125 | 5.835 | CC2 |
| s27 | 8.672 | 8.862 | 8.662 | 6.387 | 6.387 | CC1 |
| s386 | 9.676 | 9.494 | 9.311 | 6.164 | 6.164 | CC1 |
| s420 | 9.864 | 9.780 | 9.755 | 5.868 | 6.028 | CC4 |
| s510 | 9.742 | 9.742 | 9.155 | 5.324 | 5.834 | CC4 |
| s820 | 10.691 | 10.641 | 9.775 | 5.726 | 5.726 | CC1 |
| s832 | 10.975 | 10.638 | 9.866 | 6.724 | 6.401 | CC4 |

**Table 11.** *Cont.*

| Benchmark | Auto | One-Hot | JEDI | MPY | MP$_T$Y | Class of Complexity |
|:---:|:---:|:---:|:---:|:---:|:---:|:---:|
| s840 | 9.195 | 9.228 | 9.158 | 6.232 | 5.882 | CC4 |
| sand | 12.390 | 12.390 | 11.652 | 7.221 | 7.087 | CC3 |
| shiftreg | 8.302 | 7.265 | 7.091 | 5.564 | 5.564 | CC0 |
| sse | 10.096 | 9.642 | 9.455 | 5.561 | 5.561 | CC1 |
| styr | 11.067 | 11.497 | 10.666 | 5.921 | 5.719 | CC2 |
| tma | 9.831 | 10.495 | 9.821 | 5.702 | 5.596 | CC2 |
| Total | 453.73 | 454.88 | 431.08 | 267.89 | 265.88 | |
| Percentage, % | 170.65 | 171.08 | 162.13 | 100.76 | 100.00 | |

We did not obtain the values of maximum operating frequencies from Vivado reports. However, we calculated them using the values of cycle times. The frequency comparison is represented by Table 14.

As follows from Table 14, on average, the circuits of MP$_T$Y-based FSMs are faster in relation to all other models. There is the following gain: (1) 58.79% compared to Auto-based FSMs; (2) 58.7% compared to One-hot-based FSMs; (3) 61.65% compared to JEDI-based FSMs; and (4) 0.64% compared to MPY-based FSMs. Obviously, the reasons for this gain are the same as the ones discussed for the time of cycles. We will not repeat them.

Naturally, the change in the gain in frequency has the same tendencies as the change in the gain in cycle time. This statement is justified by information from Tables 15 and 16.

It should be noted that the gain in operating frequency for our method begins to appear from the complexity CC2. At the same time, the gain grows in the process of the transition to the highest categories of complexity.

Thus, if FSMs belong to the classes CC0-CC1, then equivalent MP$_T$Y and MPY FSMs have the same values of LUT counts, cycle time and maximum operating frequency. For more complex FSMs, MP$_T$Y FSMs require fewer LUTs than for equivalent MPY FSMs. In addition, for FSMs from classes CC0-CC1, both models have the same values of temporal characteristics. However, as the complexity increases, the temporal characteristics of the MP$_T$Y FSMs gradually become slightly better than they are for equivalent MPY FSMs. This gain is rather small; however, the very fact that a decrease in the number of LUTs does not lead to performance degradation is important. The results of the experiments allow us to draw the following conclusion: MP$_T$Y FSMs can replace MPY FSMs for average, big and very big sequential devices. For a more visual assessment of the results, we built a diagram (Figure 9). This diagram shows a comparison of percentages for the main characteristics of the studied methods.

To construct charts (Figure 9), we used tables in which the results are shown for all benchmarks, and not for their individual categories. To show the results for LUT counts, we used Table 7. The times of cycles are taken from Table 11. At last, the results for the values of maximum operating frequencies are derived from Table 14. It clearly follows from Figure 9 that the proposed method allows the improvement in the spatial characteristics of circuits (without the degradation of temporal characteristics).

**Table 12.** Cycle times for classes CC0-CC1 (nanoseconds).

| Benchmark | Auto | One-Hot | JEDI | MPY | MP$_T$Y | Class of Complexity |
|---|---|---|---|---|---|---|
| bbara | 8.811 | 8.811 | 8.352 | 5.214 | 5.214 | CC1 |
| bbsse | 10.096 | 9.642 | 9.213 | 5.226 | 5.226 | CC1 |
| bbtas | 8.497 | 8.497 | 8.451 | 5.308 | 5.308 | CC0 |
| beecount | 9.605 | 9.605 | 8.941 | 5.373 | 5.373 | CC1 |
| cse | 10.558 | 9.840 | 9.343 | 5.453 | 5.453 | CC1 |
| dk14 | 8.821 | 9.395 | 8.762 | 5.839 | 5.839 | CC1 |
| dk15 | 8.797 | 8.998 | 8.735 | 5.219 | 5.219 | CC1 |
| dk16 | 9.491 | 9.320 | 8.672 | 5.245 | 5.245 | CC1 |
| dk17 | 8.617 | 9.587 | 8.617 | 5.400 | 5.400 | CC0 |
| dk27 | 8.325 | 8.424 | 8.369 | 5.195 | 5.195 | CC0 |
| dk512 | 8.566 | 8.566 | 8.477 | 4.119 | 4.119 | CC0 |
| donfile | 9.033 | 9.034 | 8.509 | 5.168 | 5.168 | CC1 |
| ex2 | 8.635 | 8.635 | 8.596 | 5.624 | 5.624 | CC1 |
| ex3 | 8.731 | 8.731 | 8.707 | 5.931 | 5.931 | CC0 |
| ex4 | 9.214 | 9.315 | 8.874 | 5.481 | 5.481 | CC1 |
| ex5 | 9.147 | 9.147 | 9.119 | 5.425 | 5.425 | CC0 |
| ex6 | 9.564 | 9.772 | 9.330 | 5.369 | 5.369 | CC1 |
| ex7 | 8.598 | 8.578 | 8.584 | 5.200 | 5.200 | CC1 |
| keyb | 10.121 | 10.699 | 9.666 | 5.265 | 5.265 | CC1 |
| lion | 8.539 | 8.501 | 8.541 | 6.062 | 6.062 | CC0 |
| lion9 | 8.470 | 8.998 | 8.444 | 5.270 | 5.270 | CC0 |
| mark1 | 9.825 | 9.825 | 9.343 | 6.395 | 6.395 | CC1 |
| mc | 8.688 | 8.719 | 8.682 | 6.099 | 6.099 | CC0 |
| modulo12 | 8.302 | 8.302 | 8.299 | 5.928 | 5.928 | CC0 |
| opus | 9.684 | 9.684 | 9.275 | 5.322 | 5.322 | CC1 |
| s27 | 8.672 | 8.862 | 8.662 | 6.387 | 6.387 | CC1 |
| s386 | 9.676 | 9.494 | 9.311 | 6.164 | 6.164 | CC1 |
| s820 | 10.691 | 10.641 | 9.775 | 5.726 | 5.726 | CC1 |
| shiftreg | 8.302 | 7.265 | 7.091 | 5.564 | 5.564 | CC0 |
| sse | 10.096 | 9.642 | 9.455 | 5.561 | 5.561 | CC1 |
| Total | 274.17 | 274.53 | 264.20 | 165.53 | 165.53 | |
| Percentage, % | 165.63 | 165.85 | 159.60 | 100.00 | 100.00 | |

**Table 13.** Cycle times for classes CC2-CC4 (nanoseconds).

| Benchmark | Auto | One-Hot | JEDI | MPY | MP$_T$Y | Class of Complexity |
|-----------|------|---------|------|-----|---------|---------------------|
| ex1 | 10.425 | 10.955 | 9.454 | 5.821 | 5.741 | CC2 |
| kirkman | 10.971 | 10.392 | 10.280 | 5.612 | 5.482 | CC2 |
| planet | 11.264 | 11.264 | 9.073 | 6.018 | 5.878 | CC2 |
| planet1 | 11.264 | 11.264 | 9.073 | 6.018 | 5.834 | CC2 |
| pma | 10.634 | 10.634 | 9.681 | 6.101 | 6.101 | CC2 |
| s1 | 10.623 | 11.154 | 10.156 | 5.830 | 5.707 | CC2 |
| s1488 | 11.013 | 11.372 | 10.155 | 6.432 | 6.206 | CC2 |
| s1494 | 10.487 | 10.654 | 9.878 | 5.723 | 5.511 | CC2 |
| s1a | 10.313 | 9.462 | 9.704 | 5.689 | 5.511 | CC2 |
| s208 | 9.503 | 9.434 | 9.361 | 6.125 | 5.835 | CC2 |
| s420 | 9.864 | 9.780 | 9.755 | 5.868 | 6.028 | CC4 |
| s510 | 9.742 | 9.742 | 9.155 | 5.324 | 5.834 | CC4 |
| s832 | 10.975 | 10.638 | 9.866 | 6.724 | 6.401 | CC4 |
| s840 | 9.195 | 9.228 | 9.158 | 6.232 | 5.882 | CC4 |
| sand | 12.390 | 12.390 | 11.652 | 7.221 | 7.087 | CC3 |
| styr | 11.067 | 11.497 | 10.666 | 5.921 | 5.719 | CC2 |
| tma | 9.831 | 10.495 | 9.821 | 5.702 | 5.596 | CC2 |
| Total | 179.56 | 180.36 | 166.89 | 102.36 | 100.35 | |
| Percentage, % | 178.93 | 179.72 | 166.30 | 102.00 | 100.00 | |

**Table 14.** Experimental results (the maximum operating frequency, MHz).

| Benchmark | Auto | One-Hot | JEDI | MPY | MP$_T$Y | Class of Complexity |
|-----------|------|---------|------|-----|---------|---------------------|
| bbara | 113.496 | 113.496 | 119.727 | 191.809 | 191.809 | CC1 |
| bbsse | 99.049 | 103.713 | 108.539 | 191.342 | 191.342 | CC1 |
| bbtas | 117.687 | 117.687 | 118.336 | 188.389 | 188.389 | CC0 |
| beecount | 104.112 | 104.112 | 111.839 | 186.111 | 186.111 | CC1 |
| cse | 94.713 | 101.626 | 107.03 | 183.399 | 183.399 | CC1 |
| dk14 | 113.364 | 106.439 | 114.134 | 171.26 | 171.26 | CC1 |
| dk15 | 113.675 | 111.137 | 114.487 | 191.626 | 191.626 | CC1 |
| dk16 | 105.362 | 107.294 | 115.316 | 190.654 | 190.654 | CC1 |
| dk17 | 116.049 | 104.308 | 116.049 | 185.192 | 185.192 | CC0 |
| dk27 | 120.122 | 118.709 | 119.494 | 192.487 | 192.487 | CC0 |
| dk512 | 116.74 | 116.74 | 117.963 | 242.792 | 242.792 | CC0 |
| donfile | 110.706 | 110.696 | 117.517 | 193.504 | 193.504 | CC1 |
| ex1 | 95.922 | 91.281 | 105.777 | 171.796 | 174.19 | CC2 |
| ex2 | 115.808 | 115.808 | 116.34 | 177.799 | 177.799 | CC1 |
| ex3 | 114.536 | 114.536 | 114.846 | 168.594 | 168.594 | CC0 |
| ex4 | 108.53 | 107.352 | 112.69 | 182.443 | 182.443 | CC1 |
| ex5 | 109.327 | 109.327 | 109.661 | 184.328 | 184.328 | CC0 |
| ex6 | 104.556 | 102.333 | 107.183 | 186.268 | 186.268 | CC1 |

**Table 14.** *Cont.*

| Benchmark | Auto | One-Hot | JEDI | MPY | MP$_T$Y | Class of Complexity |
|---|---|---|---|---|---|---|
| ex7 | 116.306 | 116.576 | 116.495 | 192.304 | 192.304 | CC1 |
| keyb | 98.806 | 93.466 | 103.453 | 189.921 | 189.921 | CC1 |
| kirkman | 91.148 | 96.232 | 97.272 | 178.181 | 182.406 | CC2 |
| lion | 117.11 | 117.634 | 117.083 | 164.969 | 164.969 | CC0 |
| lion9 | 118.065 | 111.136 | 118.421 | 189.756 | 189.756 | CC0 |
| mark1 | 101.781 | 101.781 | 107.032 | 156.361 | 156.361 | CC1 |
| mc | 115.102 | 114.694 | 115.174 | 163.958 | 163.958 | CC0 |
| modulo12 | 120.454 | 120.454 | 120.498 | 168.696 | 168.696 | CC0 |
| opus | 103.265 | 103.265 | 107.818 | 187.911 | 187.911 | CC1 |
| planet | 88.777 | 88.777 | 110.222 | 166.182 | 170.14 | CC2 |
| planet1 | 88.777 | 88.777 | 110.222 | 166.159 | 171.417 | CC2 |
| pma | 94.039 | 94.039 | 103.293 | 163.902 | 163.902 | CC2 |
| s1 | 94.134 | 89.653 | 98.465 | 171.535 | 175.215 | CC2 |
| s1488 | 90.8 | 87.934 | 98.472 | 155.481 | 161.143 | CC2 |
| s1494 | 95.357 | 93.861 | 101.236 | 174.744 | 181.467 | CC2 |
| s1a | 96.963 | 105.687 | 103.048 | 175.776 | 181.467 | CC2 |
| s208 | 105.231 | 106 | 106.825 | 163.266 | 171.38 | CC2 |
| s27 | 115.314 | 112.842 | 115.449 | 156.566 | 156.566 | CC1 |
| s386 | 103.348 | 105.329 | 107.401 | 162.231 | 162.231 | CC1 |
| s420 | 101.378 | 102.249 | 102.514 | 170.42 | 165.897 | CC4 |
| s510 | 102.648 | 102.648 | 109.226 | 187.816 | 171.398 | CC4 |
| s820 | 93.537 | 93.975 | 102.3 | 174.643 | 174.643 | CC1 |
| s832 | 91.117 | 94.001 | 101.354 | 148.725 | 156.231 | CC4 |
| s840 | 108.755 | 108.364 | 109.196 | 160.471 | 170.02 | CC4 |
| sand | 80.711 | 80.711 | 85.821 | 138.478 | 141.096 | CC3 |
| shiftreg | 120.454 | 137.645 | 141.028 | 179.726 | 179.726 | CC0 |
| sse | 99.049 | 103.713 | 105.76 | 179.809 | 179.809 | CC1 |
| styr | 90.359 | 86.979 | 93.754 | 168.899 | 174.865 | CC2 |
| tma | 101.719 | 95.284 | 101.819 | 175.381 | 178.703 | CC2 |
| Total | 4918.26 | 4910.3 | 5157.58 | 8312.06 | 8365.78 | |
| Percentage, % | 58.79 | 58.7 | 61.65 | 99.36 | 100 | |

**Table 15.** Experimental results (the frequencies for classes CC0-CC1, MHz).

| Benchmark | Auto | One-Hot | JEDI | MPY | MP$_T$Y | Class of Complexity |
|---|---|---|---|---|---|---|
| bbara | 113.496 | 113.496 | 119.727 | 191.809 | 191.809 | CC1 |
| bbsse | 99.049 | 103.713 | 108.539 | 191.342 | 191.342 | CC1 |
| bbtas | 117.687 | 117.687 | 118.336 | 188.389 | 188.389 | CC0 |
| beecount | 104.112 | 104.112 | 111.839 | 186.111 | 186.111 | CC1 |
| cse | 94.713 | 101.626 | 107.030 | 183.399 | 183.399 | CC1 |
| dk14 | 113.364 | 106.439 | 114.134 | 171.260 | 171.260 | CC1 |

**Table 15.** *Cont.*

| Benchmark | Auto | One-Hot | JEDI | MPY | MP$_T$Y | Class of Complexity |
|---|---|---|---|---|---|---|
| dk15 | 113.675 | 111.137 | 114.487 | 191.626 | 191.626 | CC1 |
| dk16 | 105.362 | 107.294 | 115.316 | 190.654 | 190.654 | CC1 |
| dk17 | 116.049 | 104.308 | 116.049 | 185.192 | 185.192 | CC0 |
| dk27 | 120.122 | 118.709 | 119.494 | 192.487 | 192.487 | CC0 |
| dk512 | 116.740 | 116.740 | 117.963 | 242.792 | 242.792 | CC0 |
| donfile | 110.706 | 110.696 | 117.517 | 193.504 | 193.504 | CC1 |
| ex2 | 115.808 | 115.808 | 116.340 | 177.799 | 177.799 | CC1 |
| ex3 | 114.536 | 114.536 | 114.846 | 168.594 | 168.594 | CC0 |
| ex4 | 108.530 | 107.352 | 112.690 | 182.443 | 182.443 | CC1 |
| ex5 | 109.327 | 109.327 | 109.661 | 184.328 | 184.328 | CC0 |
| ex6 | 104.556 | 102.333 | 107.183 | 186.268 | 186.268 | CC1 |
| ex7 | 116.306 | 116.576 | 116.495 | 192.304 | 192.304 | CC1 |
| keyb | 98.806 | 93.466 | 103.453 | 189.921 | 189.921 | CC1 |
| lion | 117.110 | 117.634 | 117.083 | 164.969 | 164.969 | CC0 |
| lion9 | 118.065 | 111.136 | 118.421 | 189.756 | 189.756 | CC0 |
| mark1 | 101.781 | 101.781 | 107.032 | 156.361 | 156.361 | CC1 |
| mc | 115.102 | 114.694 | 115.174 | 163.958 | 163.958 | CC0 |
| modulo12 | 120.454 | 120.454 | 120.498 | 168.696 | 168.696 | CC0 |
| opus | 103.265 | 103.265 | 107.818 | 187.911 | 187.911 | CC1 |
| s27 | 115.314 | 112.842 | 115.449 | 156.566 | 156.566 | CC1 |
| s386 | 103.348 | 105.329 | 107.401 | 162.231 | 162.231 | CC1 |
| s820 | 93.537 | 93.975 | 102.300 | 174.643 | 174.643 | CC1 |
| shiftreg | 120.454 | 137.645 | 141.028 | 179.726 | 179.726 | CC0 |
| sse | 99.049 | 103.713 | 105.760 | 179.809 | 179.809 | CC1 |
| Total | 3300.42 | 3297.82 | 3419.06 | 5474.85 | 5474.85 | |
| Percentage, % | 60.28 | 60.24 | 62.45 | 100.00 | 100.00 | |

**Table 16.** Experimental results (the frequencies for classes CC2-CC4 MHz).

| Benchmark | Auto | One-Hot | JEDI | MPY | MP$_T$Y | Class of Complexity |
|---|---|---|---|---|---|---|
| ex1 | 95.922 | 91.281 | 105.777 | 171.796 | 174.190 | CC2 |
| kirkman | 91.148 | 96.232 | 97.272 | 178.181 | 182.406 | CC2 |
| planet | 88.777 | 88.777 | 110.222 | 166.182 | 170.140 | CC2 |
| planet1 | 88.777 | 88.777 | 110.222 | 166.159 | 171.417 | CC2 |
| pma | 94.039 | 94.039 | 103.293 | 163.902 | 163.902 | CC2 |
| s1 | 94.134 | 89.653 | 98.465 | 171.535 | 175.215 | CC2 |
| s1488 | 90.800 | 87.934 | 98.472 | 155.481 | 161.143 | CC2 |
| s1494 | 95.357 | 93.861 | 101.236 | 174.744 | 181.467 | CC2 |
| s1a | 96.963 | 105.687 | 103.048 | 175.776 | 181.467 | CC2 |
| s208 | 105.231 | 106.000 | 106.825 | 163.266 | 171.380 | CC2 |
| s420 | 101.378 | 102.249 | 102.514 | 170.420 | 165.897 | CC4 |

**Table 16.** *Cont.*

| Benchmark | Auto | One-Hot | JEDI | MPY | MP$_T$Y | Class of Complexity |
|:---:|:---:|:---:|:---:|:---:|:---:|:---:|
| s510 | 102.648 | 102.648 | 109.226 | 187.816 | 171.398 | CC4 |
| s832 | 91.117 | 94.001 | 101.354 | 148.725 | 156.231 | CC4 |
| s840 | 108.755 | 108.364 | 109.196 | 160.471 | 170.020 | CC4 |
| sand | 80.711 | 80.711 | 85.821 | 138.478 | 141.096 | CC3 |
| styr | 90.359 | 86.979 | 93.754 | 168.899 | 174.865 | CC2 |
| tma | 101.719 | 95.284 | 101.819 | 175.381 | 178.703 | CC2 |
| Total | 1617.83 | 1612.48 | 1738.51 | 2837.21 | 2890.94 | |
| Percentage, % | 55.96 | 55.78 | 60.14 | 98.14 | 100.00 | |

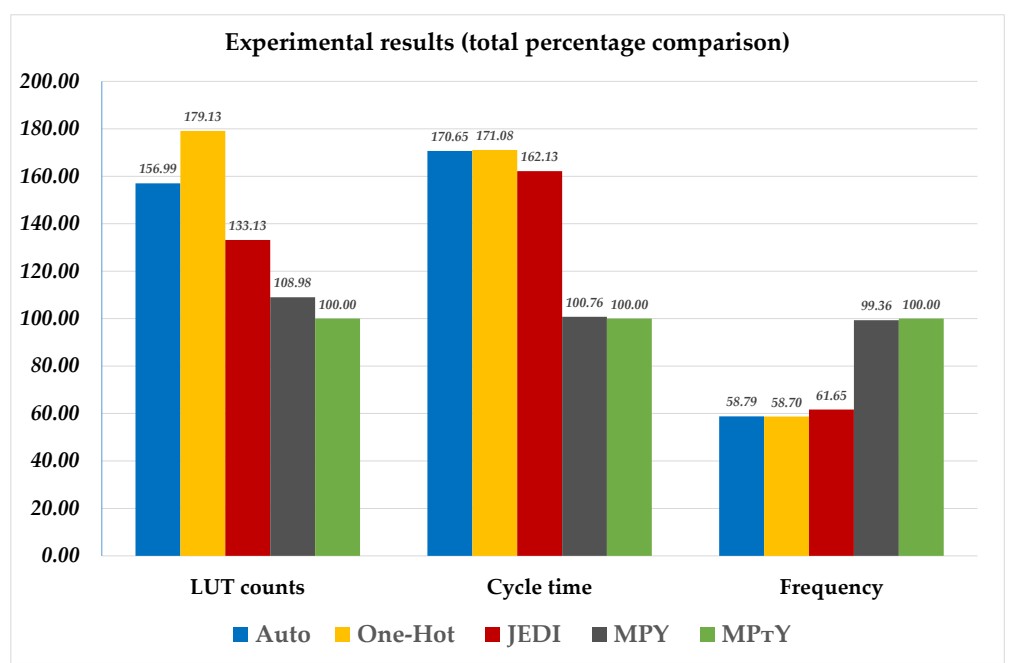

**Figure 9.** Comparison of percentages for the main characteristics of the studied methods.

## 7. Conclusions

Modern FPGAs are widely used in digital design [2]. These chips are very powerful: today, a single chip may implement a circuit with very complicated blocks [4]. Being universal, these chips have a significant drawback: they include a huge number of LUT elements with an extremely small number of inputs [3,4]. This phenomenon leads to the need to use extremely sophisticated methods for optimizing the FSM-based logic circuits. It is this shortcoming that necessitates the use of various methods of functional decomposition to obtain the resulting circuit. As a result of functional decomposition, the implemented circuits are multi-level. These circuits are slower and less energy efficient than the equivalent single-level solutions.

The use of structural decomposition methods allows the improvement in the main characteristics of multi-level FSM circuits [9]. The analysis of the work [9] leads to the conclusion that in the vast majority of cases, the SD-based FSM circuits are significantly better than their FD-based counterparts. In the paper [17], the decrease in LUT counts is achieved due to joint use of such SD-based methods as the replacement of inputs and encoding of output collections. As follows from [17], this approach allows the obtaining

of MPY FSMs, whose circuits have better characteristics compared with equivalent FD-based circuits.

To reduce the LUT count in the circuits of MPY-based FSMs, we propose to replace the maximum binary state codes with extended state codes. The proposed approach is based on using twofold state assignment [18]. As follows from the experiments, the proposed approach reduces LUT counts without the degradation of temporal characteristics as compared to equivalent MPY-based FSMs. We hope the proposed method can be used in FPGA-based designs.

**Author Contributions:** Conceptualization, A.B., L.T., M.M. and K.K.; Methodology, A.B., L.T., M.M. and K.K.; Software, A.B., L.T., M.M. and K.K.; Validation, A.B., L.T., M.M. and K.K.; Formal analysis, A.B., L.T., M.M. and K.K.; Investigation, A.B., L.T., M.M. and K.K.; Writing—original draft preparation, A.B., L.T., M.M. and K.K.; Supervision, A.B. All authors have read and agreed to the published version of the manuscript.

**Funding:** This research received no external funding.

**Data Availability Statement:** The data presented in this study are available in the article.

**Conflicts of Interest:** The authors declare no conflict of interest.

## Abbreviations

The following abbreviations are used in this manuscript:

| | |
|---|---|
| CLB | configurable logic block |
| CO | collection of outputs |
| DST | direct structure table |
| FD | functional decomposition |
| FPGA | field-programmable gate array |
| FSM | finite state machine |
| IMF | input memory function |
| LUT | look-up table |
| RG | state-code register |
| SBF | system of Boolean functions |
| SD | structural decomposition |
| STG | state-transition graph |
| STT | state-transition table |

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
