# Peer review of "Improving the Spatial Characteristics of Three-Level LUT-Based Mealy FSM Circuits"

_electronics, doi:10.3390/electronics12051133_

Round 1

Reviewer 1 Report

The article proposes an optimization for Mealy FSM circuits based on three approaches: a new architecture of LUT-based FSM circuit, uniform distribution of inputs and state encoding, and a method for stabilizing FSM outputs.

The results arevalidated using benchmarks.

Author Response

Dear Sir or Madame!

Thank you very much for your valuable time spending on thorough analysis of our article!

Yours sincerely,           

           Authors

Reviewer 2 Report

Dear authors,

Some visual graphical representation of the research outcomes should be given (graphic charts, curves, bar plots) to alleviate understanding on how you method affects the device performance in comparison with others.

line 198 - abreviation OS should be decrypted (possible typo),

line 271 - feed-back -> feedback,

line 579 -  the following phenomenon follows from (two words in one sentence with the same word root)

Author Response

Dear Reviewer,

thank you very much for your careful analysis of our article and your valuable time. We think your comments help us to improve the quality of our paper. Please, find in the attachment our answers concerning your remarks. In the original paper (pdf document) all essential changes are highlighted.

Yours sincerely,

Authors

Reviewer 3 Report

Dear authors, in my opinion, this article brings significant arguments regarding the treated subject. The research, studies and the experimental results are valuable, correctly defined and properly described. I think that your article will be interesting to readers of Electronics journal and useful for some of them.

I must note, once again, the amount of work and the involvement of the authors for the study and testing of the presented problem.

Minor problems that I see are related to the fact that the article does not follow the provided template:

All the figures must follow the same formatting: figures must be placed in an invisible column table - are the figures inserted in this way? 

All figures should be cited in the main text as Figure X instead of Fig. X. Please correct.

The text following an equation need not be a new paragraph.

Author Response

(The authors gave the same response as above.)
